# Two forms of asynchronous release with distinctive spatiotemporal dynamics in central synapses

**Gerardo Malagon[†], Jongyun Myeong[†], Vitaly A Klyachko***

Department of Cell Biology and Physiology, Washington University School of Medicine, St Louis, United States

**Abstract** Asynchronous release is a ubiquitous form of neurotransmitter release that persists for tens to hundreds of milliseconds after an action potential. How asynchronous release is organized and regulated at the synaptic active zone (AZ) remains debatable. Using nanoscale-precision imaging of individual release events in rat hippocampal synapses, we observed two spatially distinct subpopulations of asynchronous events, ~75% of which occurred inside the AZ and with a bias towards the AZ center, while ~25% occurred outside of the functionally defined AZ, that is, ectopically. The two asynchronous event subpopulations also differed from each other in temporal properties, with ectopic events occurring at significantly longer time intervals from synchronous events than the asynchronous events inside the AZ. Both forms of asynchronous release did not, to a large extent, utilize the same release sites as synchronous events. The two asynchronous event subpopulations also differ from synchronous events in some aspects of exo-endocytosis coupling, particularly in the contribution from the fast calcium-dependent endocytosis. These results identify two subpopulations of asynchronous release events with distinctive organization and spatiotemporal dynamics.

## Editor's evaluation

Spatial organization of distinct forms of release within individual synapses is a key open question in synaptic transmission. The authors provide novel insight into this question using state-of-the-art approaches to visualize asynchronous release events and their location. Their data pushes the boundaries of this form of analysis and suggests an intricate functional nano-organization within single active zones and their periphery. These findings present a novel perspective for synaptic signaling occurring at the single release site level.

**\*For correspondence:**
klyachko@wustl.edu

[†]These authors contributed equally to this work

**Competing interest:** The authors declare that no competing interests exist.

## Introduction

Evoked neurotransmitter release occurs at the synapse in several distinct forms: synchronous release that occurs within milliseconds following arrival of an action potential (AP), asynchronous release that persists for tens to hundreds of milliseconds after an AP (*Kaeser and Regehr, 2014*; *Kavalali, 2007*; *Rudolph et al., 2015*), and spontaneous release that occurs independently of AP firing (*Kavalali et al., 2011*; *Leitz and Kavalali, 2014*; *Lin et al., 2020*). Synchronous release ensures fidelity of information transmission, and numerous recent studies have focused on elucidating its nanoscale organization and regulation in central synapses (*Gramlich and Klyachko, 2019*). Asynchronous release has also been ubiquitously observed in central synapses and shown to regulate several major aspects of synaptic function, including spike timing precision, coincidence detection, and the duration of postsynaptic responses and excitability changes (*Atluri and Regehr, 1998*; *Diamond and Jahr, 1995*; *Evstratova et al., 2014*; *Goda and Stevens, 1994*; *Hefft and Jonas, 2005*; *Lu and Trussell, 2000*;

*Otsu and Murphy, 2004*; *Scheuss et al., 2007*). The spatial and temporal interrelationships between the synchronous and asynchronous release are widely believed to play critical roles in synaptic information processing, and the balance between the two components has been shown to modulate signal processing in neural circuits (*Best and Regehr, 2009*; *Deng et al., 2020*; *Hefft and Jonas, 2005*; *Luo and Südhof, 2017*; *Medrihan et al., 2015*). Yet the principles governing organization and spatiotemporal dynamics of asynchronous release at the active zone (AZ) have remained incompletely understood.

Most of what we know about asynchronous release has been revealed by electrophysiological studies, which provided important insights into the temporal properties and molecular underpinnings of this form of release (*Atluri and Regehr, 1998*; *Deng et al., 2020*; *Diamond and Jahr, 1995*; *Evstratova et al., 2014*; *Goda and Stevens, 1994*; *Luo and Südhof, 2017*; *Otsu and Murphy, 2004*; *Scheuss et al., 2007*), while spatial organization of asynchronous release across the AZ has remained enigmatic. Recently, studies using flash-and-freeze EM provided the first precise spatial information about the localization of individual asynchronous events at the AZ, revealing their distinct spatial organization from the synchronous events in hippocampal boutons (*Kusick et al., 2020*; *Li et al., 2021*). Asynchronous events were observed to preferentially occur closer to the AZ center than synchronous release events (*Kusick et al., 2020*) and to have a preferential transsynaptic alignment with postsynaptic NMDA receptors that are clustered near the center of the postsynaptic density (*Li et al., 2021*). However, this view has been challenged by a subsequent live imaging study using fluorescent glutamate sensor SF-iGluSnFR to precisely localize glutamate release from synchronous and asynchronous events. This study reached an opposite conclusion, showing that asynchronous release events are more broadly distributed across the presynaptic area than synchronous release, localize further away from the AZ center, and cover a larger area of the AZ (*Mendonça et al., 2022*). Thus the spatial organization of asynchronous release at the AZ and its interrelationship with synchronous release remains highly debatable. The wide discrepancy between these findings and the broad timescales of asynchronous release poses a question whether asynchronous release is driven by a homogeneous population of release events or combines several spatially and/or temporally distinctive release phenomena. Furthermore, whether and to what extent synchronous and asynchronous release events utilize the same or distinct release sites remains a fundamental unresolved question.

Vesicle fusion is rapidly followed by compensatory endocytosis, and several kinetically and mechanistically distinct retrieval mechanisms have been described (*Zhang et al., 2013*). These include ultrafast endocytosis with timescales of ~100–300 ms, fast endocytosis with a timescale of a few seconds, and a slow endocytosis with timescale >20 s. Some of these forms of retrieval are also characterized by differential sensitivity to calcium, with ultrafast endocytosis being largely calcium-insensitive, while fast endocytosis is calcium-sensitive (*Chanaday and Kavalali, 2018*), supporting the notion of mechanistically different retrieval pathways at the synapses. How different forms of vesicle release and retrieval are coupled with each other has been a subject of ongoing debate. Specifically, whether synchronous and asynchronous forms of vesicle release are coupled to the same or distinct forms of endocytosis remains largely unknown.

To study the organization of vesicle release, we previously developed a set of nanoscale imaging tools that permitted localization of individual release events with a precision of ~27 nm in live hippocampal synapses (*Maschi and Klyachko, 2017*; *Maschi and Klyachko, 2020*). Here we extended this approach to compare the spatiotemporal properties of synchronous and asynchronous release events in hippocampal synapses. Our results revealed the presence of two spatially and temporally distinct populations of asynchronous events, one localized within the AZ and one localized ectopically. These findings also reconcile the seemingly incompatible previous studies on the organization of asynchronous release across the AZ.

## Results
### Two spatially distinct populations of asynchronous events in hippocampal synapses

To compare the spatiotemporal properties of synchronous and asynchronous release, hippocampal neurons in culture were infected with a pH-sensitive indicator pHluorin targeted to the vesicle lumen via vGlut1 (vGlut1-pHluorin) (*Balaji and Ryan, 2007*; *Leitz and Kavalali, 2011*; *Voglmaier et al.,*

*2006*) using a lentiviral infection at DIV3, and imaged at DIV16–19 at 37°C. A continuous acquisition was performed at 50 ms/frame for 120 s, with stimulation at 1 Hz precisely synchronized with the beginning of an acquisition frame (*Figure 1Ai*). Individual AP-evoked release events were robustly detected and localized with ~27 nm localization precision throughout the observation time period (*Figure 1Aii*) as we described previously (*Maschi et al., 2018*; *Maschi and Klyachko, 2017*).

To define a population of asynchronous events, we first used temporal considerations. Asynchronous release occurs with a variable delay from an AP, ranging from tens to hundreds of milliseconds. The release events that were detected in the second frame following an AP, and thus had a delay of at least 50 ms, were considered to represent asynchronous events in our measurements. Events detected in the first frame were considered synchronous. Notably, events must occur within the first ~10–20 ms of the frame to produce enough photons to be above the detection limit, thus effectively limiting the population of events detected in the first frame to largely synchronous events. To compare the two forms of release within the same synapses, a functional representation of the AZ was first defined for each bouton. The AZ area was defined as the convolute hull of all synchronous events detected in an individual synapse and the AZ center was defined as its centroid (*Figure 1B*). We have previously found that this functional definition of the AZ dimensions closely matches the ultrastructurally defined AZ dimensions observed by EM (*Maschi and Klyachko, 2017*; *Schikorski and Stevens, 1997*). Individual release sites in each AZ were then defined by using hierarchical clustering of synchronous events with a 50 nm clustering diameter (see 'Materials and methods'; *Maschi et al., 2018*; *Maschi and Klyachko, 2017*). Subsequently, asynchronous release events detected in each bouton were examined in relationship to these synchronous event-based AZ maps.

We first used the distance to the AZ center as a simple criterion to compare the spatial properties of the two types of release, resulting in a highly significant difference in their spatial distributions (6938 synchronous events; 3891 asynchronous events, 21 coverslips from eight independent cultures; $p < 0.0001$, two-sample K-S test; *Figure 1C*; values and statistical comparisons are listed in *Supplementary file 1*), and the overall larger distance to the AZ center from asynchronous than from synchronous events (*Figure 1C*). Visual examination of the AZ maps further revealed the unexpected feature of asynchronous events evident in the presence of two spatially distinct subpopulations: one group of asynchronous events occurred within the AZ and had a tendency to localize closer to the AZ center, while the second group was localized outside of the functionally defined AZ (*Figure 1B*). To examine the spatial properties of these forms of release quantitatively, we normalized the distance measurements using a line that connects the AZ center through the event location to the AZ border (*Figure 1D*). The normalized distance of 1 represents an event occurring at the AZ border. Asynchronous events with a normalized distance to the AZ center ≤1 were considered inside the AZ ('in-AZ'), while the events with normalized distance >1 were considered outside the AZ, a subpopulation that we called ectopic events. It is worth noting that, within these definitions, all synchronous events were localized inside the AZ limits because they were used to define it (*Figure 1B*). The vast majority (98%) of synapses had asynchronous events inside the AZ, and 79% of synapses displayed ectopic asynchronous events during our observation period (*Figure 1E*). The relative preference for the 'in-AZ' subgroup varied in the synapse population in a range of 0–40% with an average of ~25% of all detected events, while for ectopic events the range was 0–20% with an average of ~9% of all events (*Figure 1E*). When comparing among asynchronous events, ~73% were localized 'in-AZ,' and ~27% were localized ectopically (*Figure 1F*). This spatial distribution of asynchronous events cannot simply arise randomly since modeled events distributed randomly in the same boutons (see 'Materials and methods') follow a significantly different distribution ($p < 0.0001$, two-sample K-S test; *Figure 1F*, *Supplementary file 1*). Moreover, in contrast to the entire population of asynchronous events, a subgroup of these events inside the AZ was localized significantly closer to the AZ center than the synchronous events in the same synapses ($p < 0.0001$, two-sample K-S test; *Figure 1G* vs. C, *Supplementary file 1*). These opposite spatial properties of the 'in-AZ' subgroup vs. the entire population of asynchronous events cannot arise from a single population with a continuum of spatial properties, which is artificially divided into two populations by our definition of the AZ limits. Moreover, the error of event localization (i.e. spatial precision of localization) was nearly the same for synchronous and asynchronous release events either inside or outside of the AZ (*Figure 1—figure supplement 1A*), and thus cannot account for the observed differences in their spatial distribution.

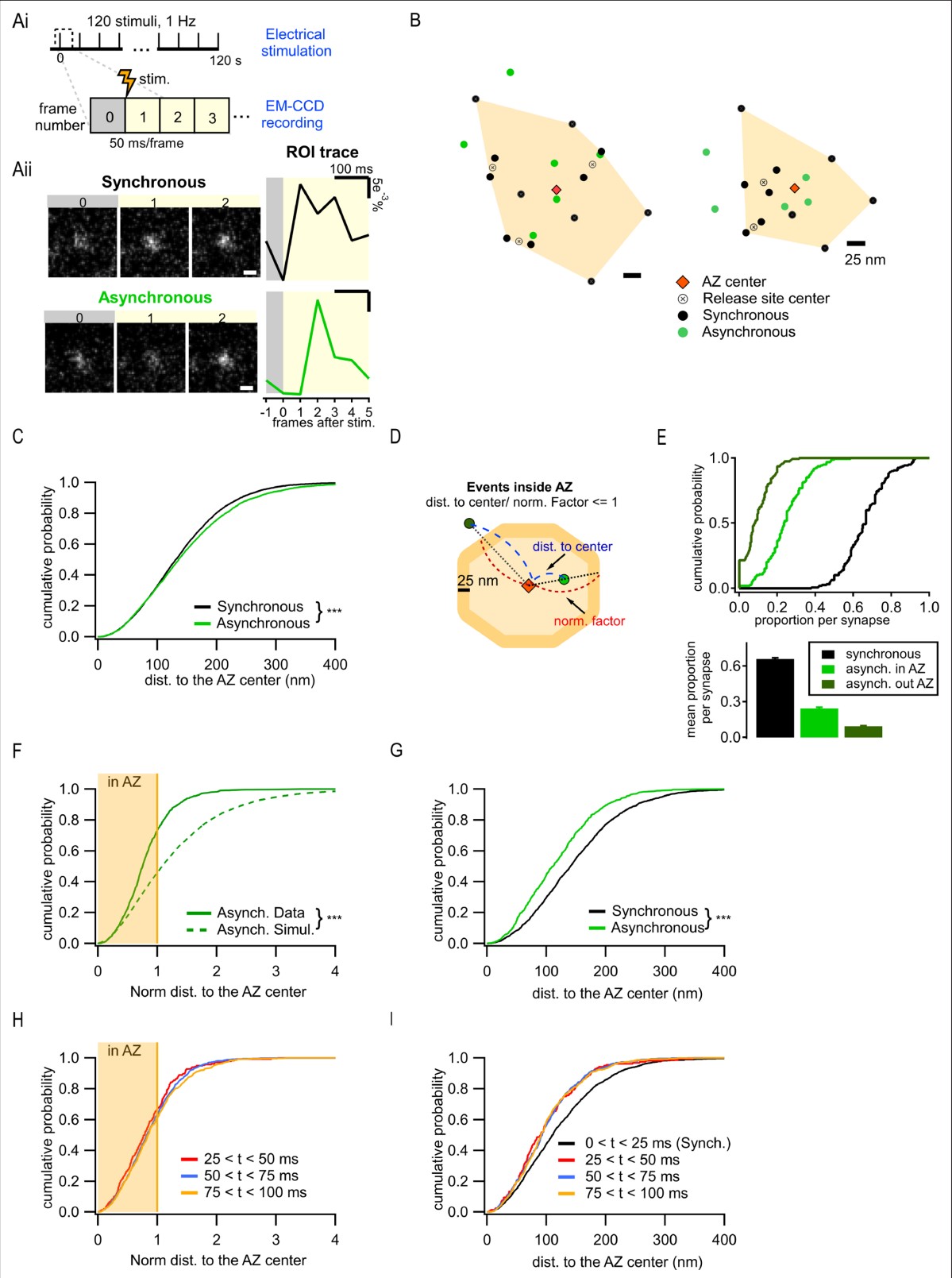

**Figure 1.** Two spatially distinct populations of asynchronous release events at individual synaptic boutons. (**Ai**) Stimulation and recording protocol. Each square represents a single frame of 50 ms duration. (**Aii**) Left: examples of raw fluorescence images of a single vesicle release event before (frame 0) and after (frames 1,2) a single action potential (AP) stimulation, for one synchronous (top) and one asynchronous (bottom) event. Scale Bar is 500 nm. Right: the corresponding event time course was determined with a circular ROI of 10 pixels and expressed as dF/F0. Note the frame number in

*Figure 1 continued on next page*

*Figure 1 continued*

which an increase in fluorescence is detected: #1 for synchronous and #2 for asynchronous. (**B**) Two examples of active zones (AZs) with the localization of synchronous (black) and asynchronous (green) events. The AZ areas are delimited by the convex hull of the synchronous events (orange), whose centers are depicted by an orange diamond. The locations of release sites (crossed circles) were determined by hierarchical clustering analysis of synchronous events. (**C**) Cumulative distribution of the distances to the AZ center for synchronous (black) and asynchronous (green) events (***p<0.0001, K-S test). (**D**) Cartoon illustrating the normalization of the distance to the AZ center for release events. The orange area represents an AZ, with its center (red diamond) and two examples of release events, one localized inside the AZ area (light green) and one outside (dark green). The 25 nm rim (approximating vesicle radius) was added to the AZ border to account for the uncertainty in the localization of the events that define the AZ area. Distance normalization was performed using the line connecting the AZ center and the event's location, such that the distance from the AZ center to the event (delimited in blue) was normalized to the distance from the center to the border of the AZ (delimited in red) along the same line. Events with normalized distance to the AZ center ≤1 were classified as being inside the AZ; events with normalized distance >1 were classified as being outside the AZ. (**E**) Cumulative distributions (top) and mean proportions per synapse (bottom) of synchronous events, and asynchronous events inside (light green) or outside (dark green) the AZ (as defined in **D**) across the synapse population. (**F**) Cumulative distributions of the normalized distances to the AZ center for all asynchronous events (continuous green line, N = 1089) and for simulated asynchronous events whose localization was randomly assigned (see 'Materials and methods') (dotted green line, N = 10,890) (***p<0.0001, K-S test). The shaded area represents the fraction of events localized inside the AZ (normalized distance to the center ≤1). (**G**) Cumulative distributions of the distances to the AZ center for a subpopulation of asynchronous events localized inside the AZ (green line, N = 799) and for synchronous events (black line, N = 1956) (***p<0.0001, K-S test). (**H**) Cumulative distributions of the normalized distances to the AZ center for asynchronous events recorded with 25 ms/frame acquisition rate using near-TIRF imaging and detected in three temporal windows: 25–50 ms (red), 50–75 ms (blue), and 75–100 ms (orange). The shaded area represents the fraction of events localized inside the AZ (normalized distance to the center ≤1). (**I**) Cumulative distributions of the distances to the AZ center for events recorded with 25 ms/frame acquisition rate using near-TIRF imaging, comparing synchronous events (black, 0–25 ms) and asynchronous events detected in three temporal windows: 25–50 ms (red), 50–75 ms (blue), and 75–100 ms (orange).

The online version of this article includes the following figure supplement(s) for figure 1:

**Figure supplement 1.** Ectopic asynchronous events are not an artifact of the variable localization precision or insufficient number of detected events.

We note that the above analyses rely on the functional definition of the AZ limits. Our current definition is in close agreement with the ultrastructural measurements of the AZ dimensions in these synapses (*Maschi and Klyachko, 2017*; *Schikorski and Stevens, 1997*), and with the number of detected event clusters/release sites per AZ (*Figure 1—figure supplement 1B and C*; *Sakamoto et al., 2018*; *Tang et al., 2016*). Nevertheless, this definition depends on our ability to fully 'sample' the available release sites within each AZ during the observation period. We thus asked whether ectopic events could simply represent locations of release sites that were not 'sampled' by the synchronous events and thus mistakenly appear as being outside of the functional AZ limits. If this is the case, then the proportion of asynchronous events inside the AZ should asymptotically approach 100% as the number of detected synchronous events per synapse increases. In contrast, we found that this proportion reached an asymptote of ~83% (*Figure 1—figure supplement 1D*), indicating that ~17% of asynchronous release still occurs outside of the functionally defined AZ limits. Reaching the asymptote (tau of approximately six events, *Figure 1—figure supplement 1D*) implies that once a certain number of synchronous events has been detected, detecting more events does not alter the proportion of asynchronous events that are inside/outside the AZ. Similarly, the mean normalized distance to the AZ center per synapse reaches the asymptote following an exponential with a tau of approximately five events (*Figure 1—figure supplement 1E*). These analyses provide further support for the presence of two distinct populations of asynchronous release events, one localized inside the AZ and another localized ectopically.

In the subsequent spatial analyses, to achieve a compromise between reducing this sampling issue and keeping enough data to perform the required analyses, we considered only boutons with a minimum of 10 detected events during the observation period. Placing the cutoff at 15 events during a 120 s period would drastically reduce the dataset, making most of the analyses impossible.

The 50 ms acquisition time per frame used in our measurements is required for our imaging approach to achieve adequate signal-to-noise ratio (SNR) for the precise event localization. Although ectopic asynchronous events represent a relatively small proportion of all detected events (~9%), it is possible that some ectopic events may be detected in the first frame and thus misidentified as synchronous events. This would broaden the definition of the AZ dimensions and shift the apparent occurrence of asynchronous events to less ectopic and more 'in-AZ' events. To test whether the limitations placed by the duration of acquisition contribute to the observed spatial differences among various forms of release, we sought to shorten the acquisition time from 50 ms to 25 ms by taking advantage of

the near-total internal reflection fluorescence (near-TIRF) imaging (*Myeong and Klyachko, 2022*). Accordingly, this improved temporal resolution shortened the definition of synchronous release to within 25 ms from the stimulus and allowed us to compare it to three temporal windows for asynchronous release (25–50 ms, 50–75 ms, and 75–100 ms). We observed essentially the same results as above, including the presence of ~35% of ectopic release among asynchronous events (*Figure 1H*), and the preferential bias of asynchronous events inside the AZ for the AZ center (*Figure 1I*). We also found that the properties of asynchronous release were very similar for the three temporal windows examined, with a slight increase in the proportion of ectopic vs. 'in-AZ' asynchronous events with time, from 33% to 37% to 39% for the three time windows, respectively. This result suggests that ectopic events are more likely to occur with longer time delays (explored in more details below). The increased proportion of ectopic events (~35%) in these measurements as compared to 50 ms/frame recordings (~27%) is consistent with the notion that the AZ dimensions are slightly broadened in 50 ms recordings. Overall, this analysis demonstrates that the precise duration of the temporal windows used to define synchronous vs. asynchronous release does not affect the observed core differences in the spatial organization of the two forms of release, at least at the timescales tested. Because SNR was

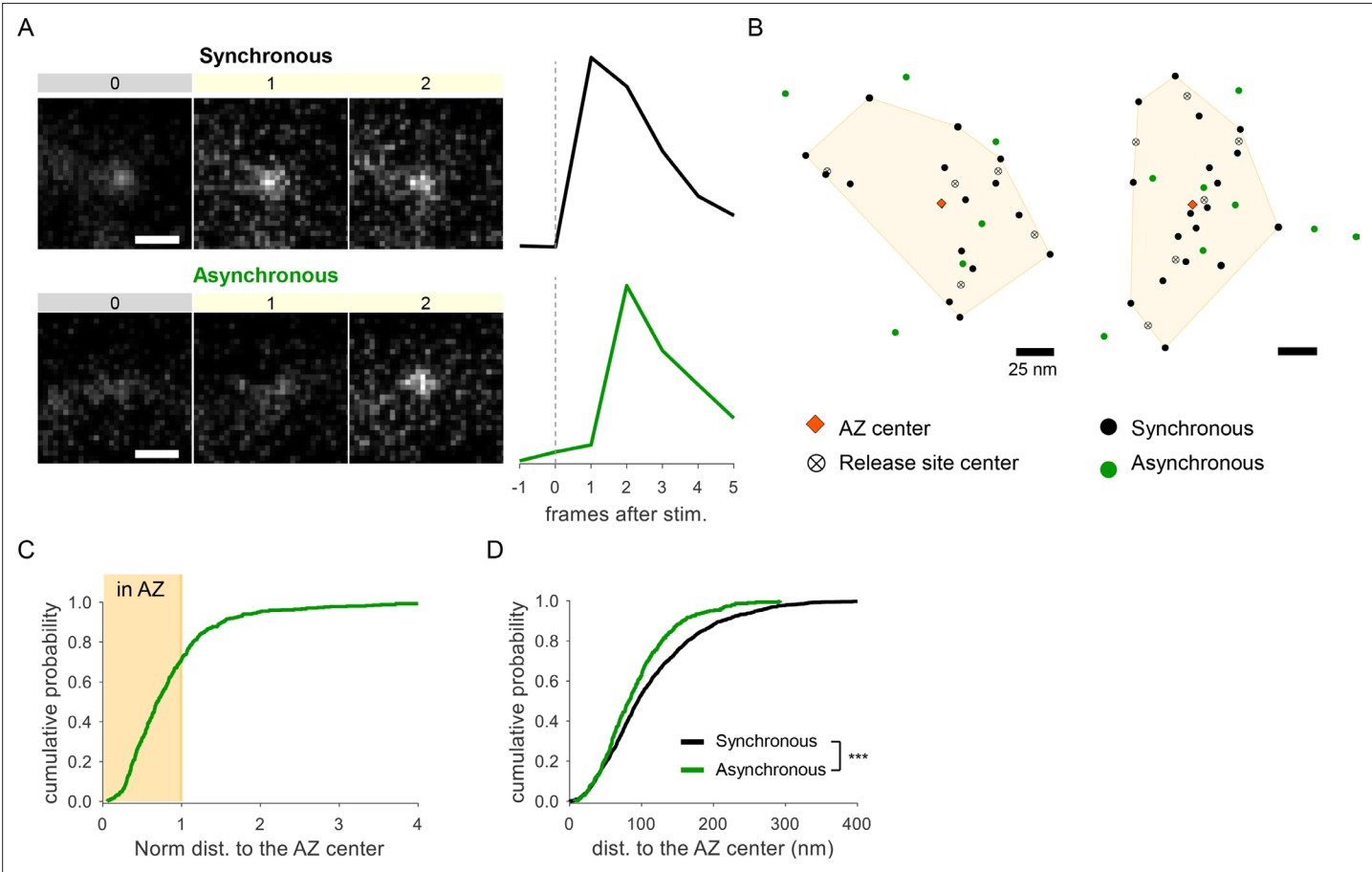

**Figure 2.** Detection of two spatially distinct populations of asynchronous events at individual synaptic boutons using glutamate sensor iGluSnFR. (**A**) Examples of raw fluorescence images of single glutamate release events detected using SF-iGluSnFR(A184S) before (frame 0) and after (frames 1,2) a single action potential (AP) stimulation, for one synchronous (top) and one asynchronous (bottom) event. Scale Bar is 1 μm. Each square represents a single frame of 50 ms duration. Right: the corresponding event time course was determined with a circular ROI of 10 pixels and expressed as dF/F0. (**B**) Two examples of active zones (AZs) with the localization of synchronous (black) and asynchronous (green) events detected by imaging of SF-iGluSnFR(A184S). The AZ areas are delimited by the convex hull of the synchronous events (yellow), whose centers are depicted by an orange diamond. The locations of release sites (crossed circles) were determined by hierarchical clustering analysis of synchronous events. (**C**) Cumulative distribution of the normalized distances to the AZ center for all asynchronous events detected by near-TIRF imaging of SF-iGluSnFR(A184S). The shaded area represents the fraction of events localized inside the functionally defined AZ (normalized distance to the center ≤1). 1370 asynchronous events from 105 synapses, 10 coverslips, 3 independent cultures. (**D**) Cumulative distributions of the distances to the AZ center for a subpopulation of asynchronous events localized inside the AZ (green line) and for synchronous events (black line) as detected by imaging of SF-iGluSnFR(A184S).

increased and the number of detected events decreased in recordings with 25 ms/frame acquisition time, subsequent experiments were performed using 50 ms acquisition time.

To confirm our observations using an independent approach, we performed detection of glutamate release events at single synapses using glutamate sensor SF-iGluSnFR(A184S) (*Marvin et al., 2018*), in combination with a near-TIRF imaging (see 'Materials and methods'). Under the same experimental conditions, stimulation protocol, and definitions used above, this approach provided a robust detection of both synchronous and asynchronous individual glutamate release events (*Figure 2A*). In close agreement with the above results, we observed two populations of asynchronous events: ~71% of which were localized inside the AZ, while ~29% were localized ectopically (*Figure 2B and C*). Moreover, the subpopulation of asynchronous events inside the AZ had a strong spatial bias toward the AZ center (*Figure 2D*), further supporting the above results. Thus independent measurements support the presence of two spatially distinct populations of asynchronous events. We note that SF-iGluSnFR is localized throughout the plasma membrane and responds to glutamate diffusing from the site of vesicle fusion, resulting in a broader signal profile than produced by vGlut1-pHluorin, which response is localized to a single synaptic vesicle undergoing fusion. Thus measurements with SF-iGluSnFR have an intrinsically lower localization precision (by approximately threefold) comparing to vGlut1-pHluorin. Because the focus of this study is on precise spatial organization of different forms of release, all subsequent analyses were performed using vGlut1-pHluorin imaging.

## Differential utilization of release sites by synchronous and asynchronous release events

Another key spatial measure is localization of asynchronous events relative to the release sites utilized by synchronous events. This measure is independent of a specific definition of the AZ dimensions and allows us to approach a fundamental question: do asynchronous and synchronous events utilize the same or distinct release sites? Clustering of synchronous events using a hierarchical clustering algorithm with 50 nm diameter (see 'Materials and methods') revealed 8.9 ± 0.2 clusters/release sites per AZ (*Figure 1—figure supplement 1B*) or 2.8 ± 0.1 repeatedly reused release sites per AZ (with two or more events detected) (*Figure 1—figure supplement 1C*), in close agreement with the previous estimates (*Gramlich and Klyachko, 2019*; *Maschi and Klyachko, 2017*; *Sakamoto et al., 2018*; *Tang et al., 2016*). We then determined the distance from every asynchronous event to the center of the closest release site. Given the cluster diameter of 50 nm used in our analyses to define release sites, and a typical vesicle radius of ~25 nm, asynchronous events within 25 nm from the center of a release site were considered to occur in that site (*Figure 3A*). This analysis showed that, for the subpopulation of asynchronous events localized inside the AZ, only 25.4% occurred at release sites defined by synchronous events (*Figure 3B*). This small subpopulation was not significantly different in their distances to the AZ center from asynchronous events that did not utilize the same release sites, although the latter show a tendency to be slightly closer to the center (p=0.053, two-sample K-S test; *Figure 3C*, *Supplementary file 1*). Considering that ~27% of asynchronous release events are ectopic and by definition do not utilize the release sites involved in synchronous release, the overall overlap of two forms of release in the same release sites is only ~15%. To verify these observations, we used an alternative, nearest-neighbor analysis (*Maschi and Klyachko, 2017*) to quantify the distance from every asynchronous event inside the AZ to the nearest synchronous event (*Figure 3D*). To keep the two analyses comparable, this quantification was limited to synchronous events corresponding to release sites undergoing repeated utilization (with at least two release events detected). Similar to the clustering analysis above, we observed that only ~20% of asynchronous events were localized within 50 nm from a synchronous event (*Figure 3D*), supporting our findings above. We note that because of the limited duration of our recordings necessitated by the natural displacement of synapses in culture (*Maschi and Klyachko, 2017*), we cannot exclude the possibility that a somewhat larger degree of spatial overlap between synchronous and asynchronous release may exist. In summary, these results suggest a distinctive utilization of release sites by synchronous and asynchronous release events.

## Distinct temporal features of the two populations of asynchronous release events

In addition to observing two spatially distinct subpopulations of asynchronous events, we asked whether there are temporal differences between these two groups. To approach this question, we

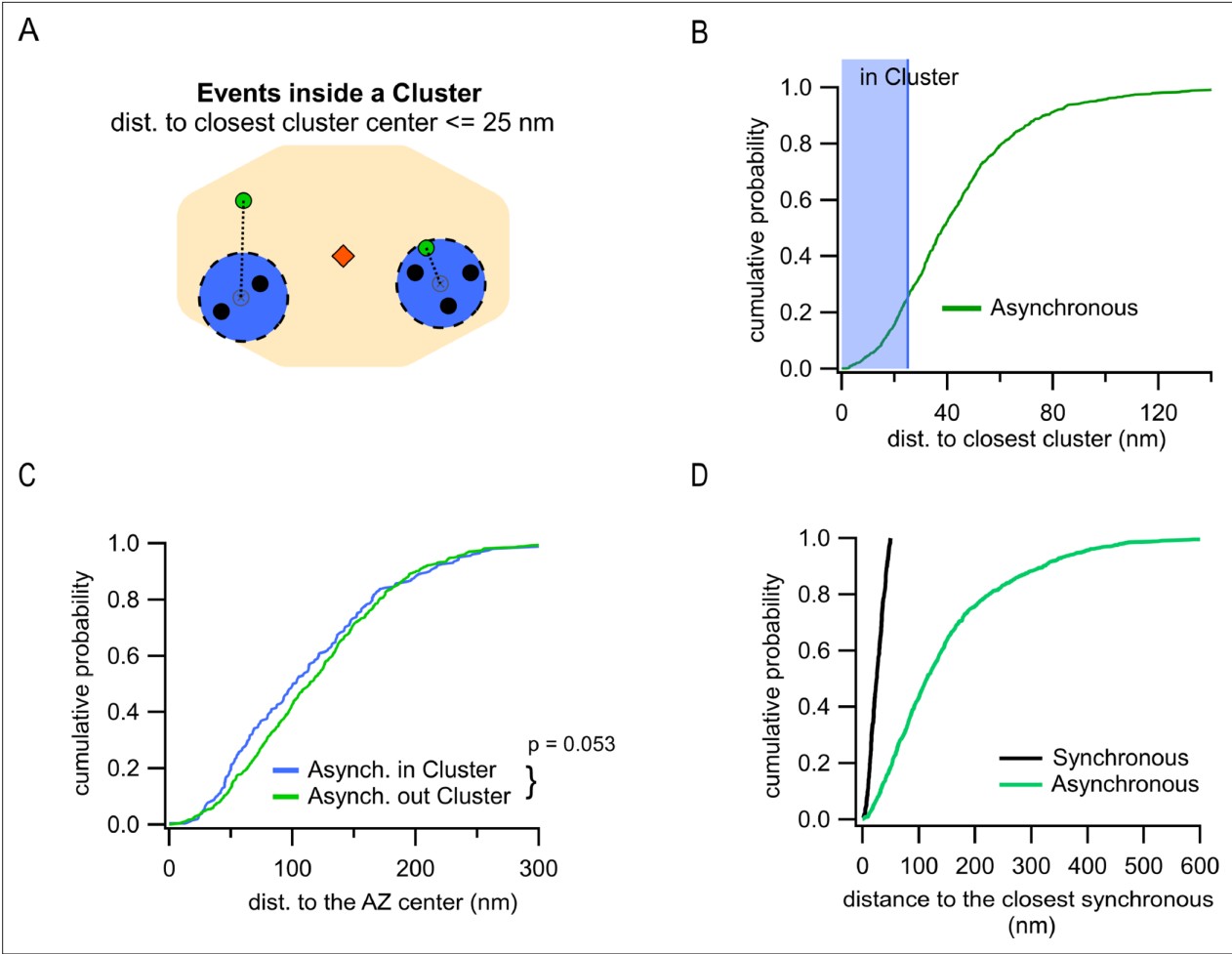

**Figure 3.** Differential utilization of release sites by synchronous and asynchronous release events. (**A**) Cartoon showing a single active zone (AZ) (orange) with two sample clusters/release sites (blue circles) illustrating the assignment of asynchronous events (green dots) as being inside or outside a cluster. Black dots represent synchronous events inside a cluster. Asynchronous events localized ≤25 nm from any cluster center were considered to be a part of that cluster (right example), while asynchronous events at longer distances to any cluster center were considered as being outside of a cluster (left example). (**B**) Cumulative distribution of the distances to the closest cluster for the subpopulation of asynchronous events inside the AZ (N = 799). The shaded area highlights the proportion of events that were considered as part of a cluster. (**C**) Cumulative distributions of the distances to the AZ center for asynchronous events within the AZ, which are classified as being either inside (blue, N = 203) or outside (green, N = 596) a cluster (n.s., p=0.053, K-S test). (**D**) Cumulative distributions of the distances from synchronous or asynchronous events to the nearest synchronous event. Only synchronous events corresponding to release sites undergoing repeated utilization (with at least two release events detected) were used in this analysis.

used the timing of synchronous release events as temporal points of reference; for every asynchronous event we measured the time elapsed from and to, respectively, the immediately preceding and immediately following synchronous events. We found that the ectopic asynchronous events display significantly longer time intervals from either the preceding synchronous events (*Figure 4A*) or the subsequent synchronous events (*Figure 4B*) compared to asynchronous events within the AZ. When only the subpopulation of asynchronous events inside the AZ was considered, we did not observe temporal differences for events detected inside and outside of release sites (*Figure 4C and D*). These results suggest that the ectopic asynchronous events are isolated not only spatially but also temporally from their counterparts inside the AZ. These findings support the presence of two distinct populations of asynchronous release events with different temporal and spatial features.

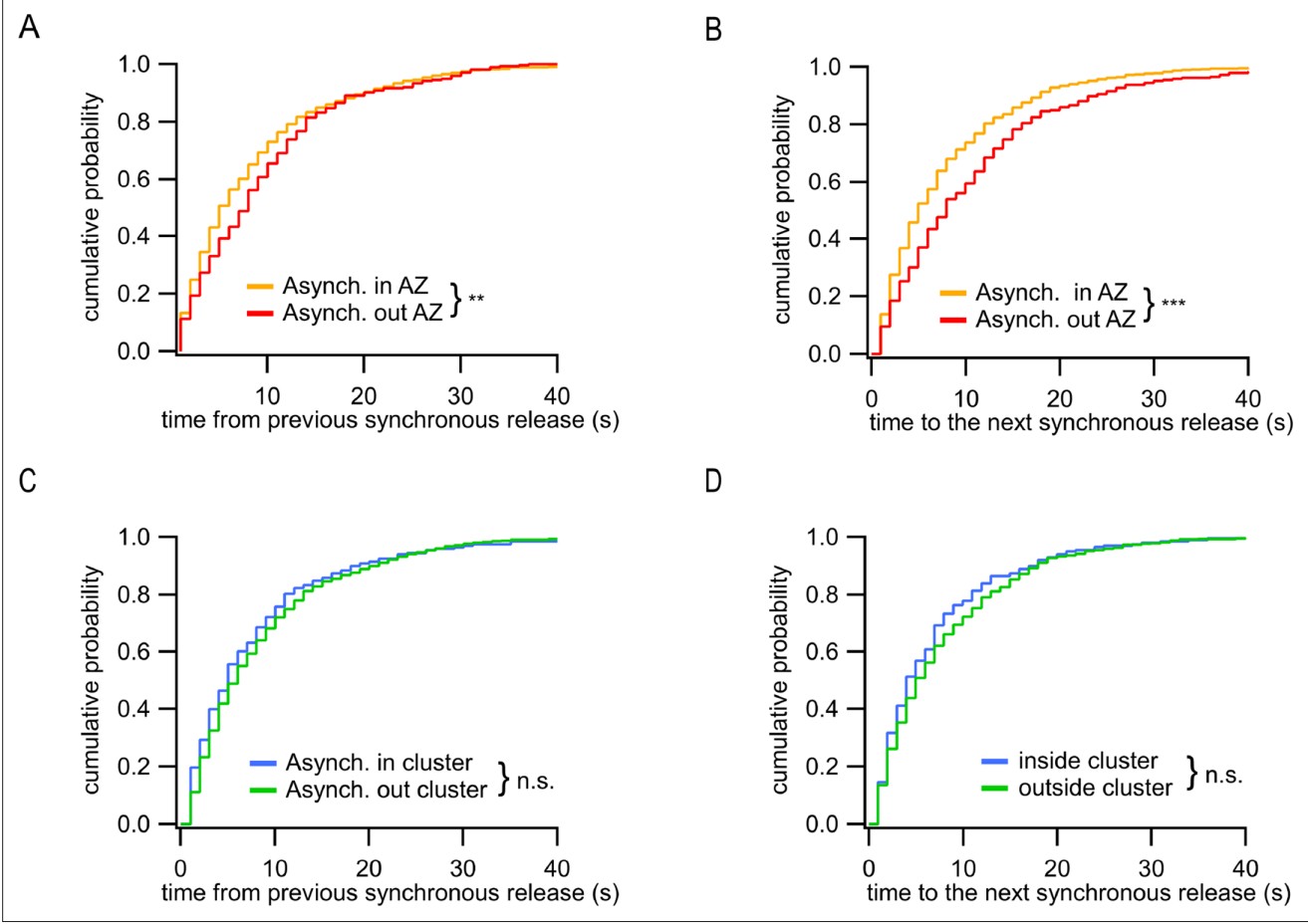

**Figure 4.** Distinct temporal features of ectopic asynchronous release events. (**A**) Cumulative distributions of the time period between each asynchronous event and the immediately preceding synchronous event for asynchronous events inside the active zone (AZ) (yellow, N = 799) or outside the AZ (red, N = 290) (\*\*p=0.0017, K-S test). (**B**) Same as (**A**) for the time period from the two types of asynchronous events to the next synchronous event (\*\*\*p<0.0001, K-S test). (**C**) Same as (**A**) for asynchronous events inside a cluster (blue, N = 203) or outside a cluster (green, N = 596) (n.s., p=0.18, K-S test). (**D**) Same as (**B**) for asynchronous events inside a cluster (blue, N = 203) or outside a cluster (green, N = 596) (n.s., p=0.31, K-S test).

## Differences in exo-endocytosis coupling of synchronous and asynchronous events

Vesicle exocytosis is rapidly followed by endocytosis, and several mechanistically and kinetically different forms of vesicle retrieval have been identified (*Chanaday and Kavalali, 2018*). Whether different forms of vesicle release are coupled with different forms of retrieval remains poorly understood. Interestingly, at least in the case of synchronous release, fusion of q-dot-labeled synaptic vesicles is preferentially coupled with different forms of endocytosis depending on the spatial location of vesicle release (*Park et al., 2012*). Given distinctive spatial organization of synchronous and asynchronous release events, we asked whether these two forms of release utilize different exo-endocytosis coupling mechanisms.

The kinetics of the pHluorin signal following vesicle fusion is widely believed to reflect, in a large part, particularities of the exo-endocytosis processes (*Balaji and Ryan, 2007*; *Chanaday and Kavalali, 2018*; *Leitz and Kavalali, 2011*; *Leitz and Kavalali, 2014*). As the noisy nature of the individual events hindered their precise kinetic comparison within a single synapse, we first compared the average fluorescence traces of synchronous and asynchronous events across the synapse population (*Figure 5A*). The fluorescence decay of the average release events had two clearly distinct kinetic components (~60 ms and ~800 ms) (*Figure 5B*), in close agreement with previous studies identifying these two components as ultrafast and fast endocytosis, respectively (*Balaji and Ryan, 2007*; *Chanaday and Kavalali, 2018*; *Leitz and Kavalali, 2011*; *Leitz and Kavalali, 2014*). For synchronous events, ultrafast and fast

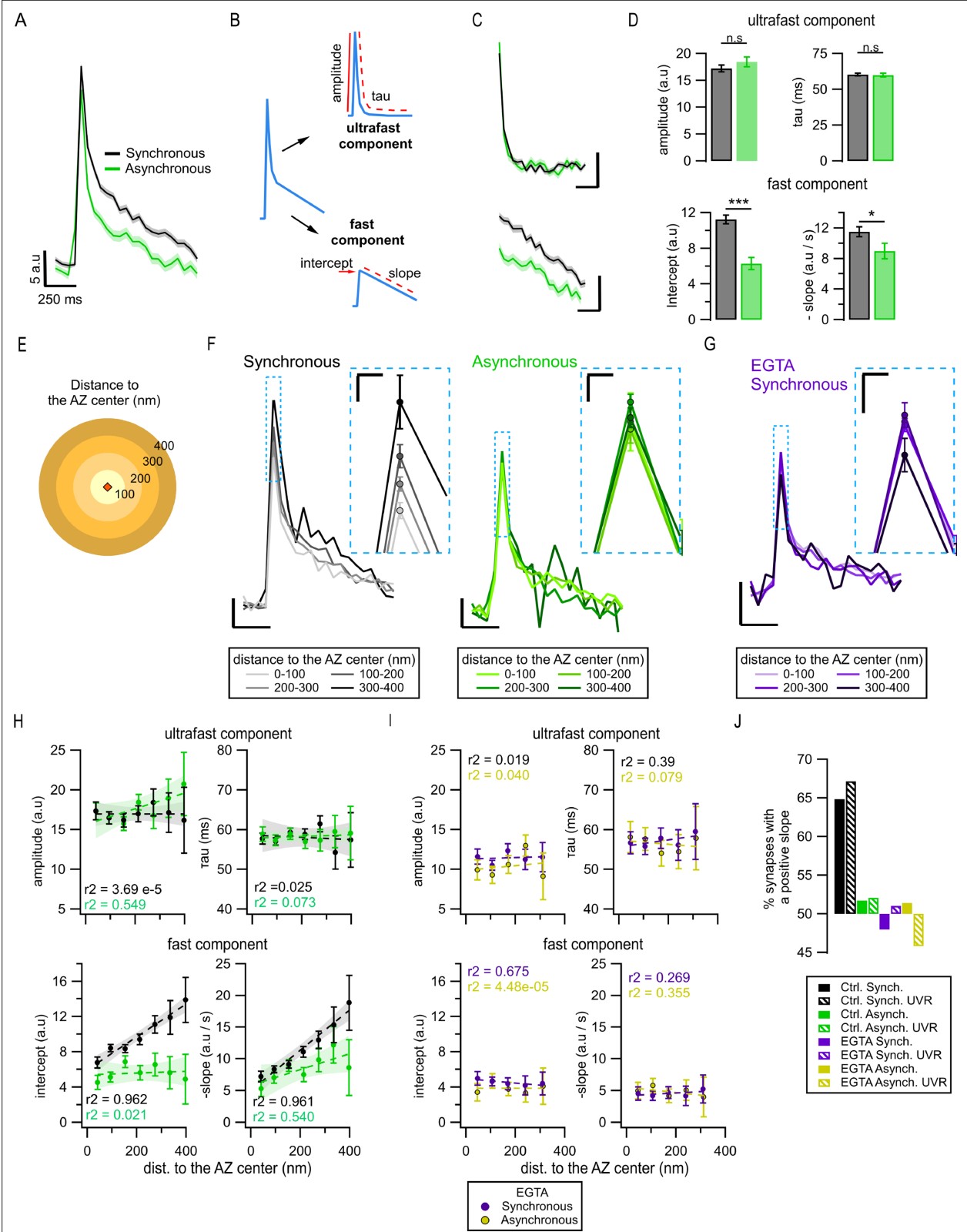

**Figure 5.** Exo-endocytosis coupling of synchronous and asynchronous release events. (**A**) Mean pHluorin signals of synchronous (black, N = 1774) and asynchronous (green, N = 975) release events; SEM displayed as a shading. (**B**) Cartoon illustrating how the decay of the pHluorin signal was dissected in two components for each individual event (see 'Materials and methods'): the first, known as *ultrafast* component, was characterized by the tau and amplitude of an exponential decay; the second, known as *fast*, was defined by the intercept and slope of a linear fit. (**C**) Average ultrafast

*Figure 5 continued on next page*

*Figure 5 continued*

and fast components from the traces in (**A**). Each component was isolated from the traces of individual events and then averaged. Calibration bars: 5 a.u., 250 ms. (**D**) Quantification of ultrafast (top) and fast (bottom) components from the individual synchronous (black) and asynchronous (green) events (*p=0.028; ***p<0.0001, n.s., not significant, K-S test or two-sample *t*-test). (**E**) Cartoon representing the concentric spatial bins (100 nm width) around the active zone (AZ) center (red diamond), used to average the pHluorin signals at the individual boutons. (**F**) Mean pHluorin signals per bin (as in **E**), for synchronous (black) and asynchronous (green) release events. The inset is a zoom-in on the peak of the signal showing increase in amplitude with the distance to the AZ center for synchronous, but not for asynchronous events. Calibration bars: 5 a.u., 250 ms. Inset: 2.5 a.u., 25 ms. (**G**) Mean pHluorin signals per bin for synchronous events in the presence of EGTA-AM. The inset is a zoom-in of the peak showing that the signal's amplitude no longer increases with the distance to the AZ center. Calibration bars: 5 a.u., 250 ms. Inset: 2.5 a.u., 25 ms. (**H**) Quantification of the ultrafast (top) and fast (bottom) components of the pHluorin signal decay as a function of the distance to the AZ center for synchronous (black) and asynchronous (green) events. The circles represent the binned data with SEM bars. The lines are linear fits of the data, and the shadows are the confidence intervals for the linear regression of the data. (**I**) Same as in (**H**) but for the events recorded in the presence of EGTA-AM. Note the lack of correlation between the distance to the AZ center and any of the features measured. (**J**) Proportion of synapses that show a positive correlation between the amplitude of the pHluorin signal and the distance to the AZ center (see 'Materials and methods'), for the two forms of release and their different treatments. Bars labeled as univesicular (UVR) correspond to the analyses in which we removed multivesicular (MVR) events (i.e. UVR-only conditions). The values are plotted relative to 50%, which represents the probability of observing a positive or negative correlation by chance. Note that EGTA-AM strongly reduces the proportion of synapses with a positive correlation for synchronous release, while removing MVR events does the opposite.

The online version of this article includes the following figure supplement(s) for figure 5:

**Figure supplement 1.** Exo-/endocytosis coupling of different forms of release.

**Figure supplement 2.** Spatial dependence of endocytosis kinetics of different forms of release.

**Figure supplement 3.** Differential exo-/endocytosis coupling of synchronous and asynchronous release is not caused by differences in localization precision or lateral diffusion.

---

components represented ~58% and ~42% of signal decay, respectively, while for asynchronous events this proportion was shifted toward smaller relative contribution of the fast component (*Figure 5—figure supplement 1A*). Further quantification of the average synchronous vs asynchronous event traces revealed significantly larger average synchronous event amplitude (p<0.001, *Figure 5—figure supplement 1H*), and major differences in the fast component of decay, which was significantly larger for synchronous than for asynchronous events (*Figure 5C and D*). In contrast, we found no significant differences in ultrafast component of endocytosis between the two types of release (*Figure 5C and D*). There were also no differences observed in endocytosis kinetics between the two subpopulations of asynchronous events, 'in-AZ' vs. ectopic (*Figure 5—figure supplement 1B and C*). We note that a larger amplitude of a decay component represents less efficient endocytosis since more of the fluorescence signal remains to be endocytosed.

We next explored whether the observed differences in amplitude and endocytosis kinetics between the two forms of release may arise from their different spatial organization. To approach this question, we grouped release events as a function of distance from the AZ center using concentric rings of 100 nm in width (*Figure 5E*) and averaging events detected inside each ring. Surprisingly, we observed a significant increase in the amplitude of synchronous events with distance from the center of the AZ toward periphery (*Figure 5F* and *Figure 5—figure supplement 1D*). In contrast, asynchronous events in the same synapses did not show any measurable differences in amplitude across the AZ (*Figure 5F* and *Figure 5—figure supplement 1D*). Importantly, the distance-dependent increase in synchronous event amplitude was largely abolished by pre-incubation with 100 µM EGTA-AM (*Figure 5G* and *Figure 5—figure supplement 1E*), suggesting that it is a calcium- and release-type-dependent phenomenon. We quantified the correlation between event amplitude and distance to the AZ center using Pearson's linear correlation algorithm. We observed a strong (0.98) and highly significant (p=0.015) correlation for synchronous release, while there was no significant correlation found for asynchronous release (p=0.92; *Figure 5—figure supplement 1D*), or in the presence of EGTA-AM (p=0.25; *Figure 5—figure supplement 1E*).

To support these observations, we used the spectral analysis of the traces, an approach that offers an advantage of being model-free as it does not rely on any fitting. We compared the average spectral composition of events occurring near the AZ center (0–100 nm ring) vs. events localized toward the AZ periphery (300–400 nm ring). In line with the above observations, synchronous events showed significant differences in spectral composition between the two distances, while asynchronous events did not (*Figure 5—figure supplement 1F and G*). Moreover, the major difference between the average

spectral composition of synchronous and asynchronous event traces was in the frequency range of 0.5–4 Hz, while no differences were evident above ~5 Hz (*Figure 5—figure supplement 1F and G*). This implies the differences in the decay component in the range of hundreds of milliseconds to a few seconds, and a similarity in the fastest decay component, thus supporting the above results.

To understand the basis of this distance-dependent phenomenon and the differences between the two forms of release, we considered that the averaged peak amplitude of single fusion events could be influenced by several factors, including the effectiveness of endocytosis, the form of vesicle release, particularly the proportion of univesicular (UVR) and multivesicular (MVR) events (*Leitz and Kavalali, 2011*; *Maschi and Klyachko, 2020*), the diffusion of vesicular components, and differences in localization precision due to variable timing of vesicle fusion. To examine the first possibility, we recorded vesicle release events in the presence of a potent dynamin inhibitor dyngo-4a (50 μm). Application of dyngo-4a resulted in a markedly reduced number of detected events, precluding spatial analyses of this dataset. Nevertheless, we observed that dyngo-4a eliminated the differences in average event amplitude between synchronous and asynchronous release (*Figure 5—figure supplement 1H*), supporting the hypothesis that these amplitude differences are endocytosis-dependent. To further examine this possibility, we quantified the two components of event decay as a function of the distance to the AZ center in normal conditions. For synchronous events, we found an increase in the fast component of decay with the distance from the AZ center (*Figure 5H*, bottom), which closely paralleled the increase in the event amplitude (*Figure 5F* and *Figure 5—figure supplement 1D*), while the ultrafast component of decay remained unaltered across the AZ (*Figure 5H*, top). Notably, this distance dependence of the fast component of decay was eliminated by pre-incubation with EGTA-AM (*Figure 5I*), which parallels the effect of EGTA-AM on the event amplitude (*Figure 5G* and *Figure 5—figure supplement 1E*). In contrast, for asynchronous release events, both ultrafast and fast components of decay were homogeneous across the AZ and did not correlate with the distance to AZ center (*Figure 5H*). Moreover, as was the case with the event amplitude, both ultrafast and fast components of decay were no longer different between synchronous and asynchronous events in the presence of EGTA-AM (*Figure 5I*).

We next examined whether variable proportion of MVR/UVR contributes to the distance-dependent differences between the two forms of release. If the proportion of synchronous MVR events increases with distance from the AZ center, it can cause an apparent increase in the average synchronous event amplitude. If this is the case, then we predict that excluding MVR events from the analysis should reduce or eliminate the observed distance dependence of synchronous, but not asynchronous events. We examined this possibility by performing detection of MVR events, which were identified as events with a normalized amplitude larger than 2 (see 'Materials and methods'; *Maschi et al., 2021*; *Maschi and Klyachko, 2020*), and excluding them from this analysis. In contrast with the above prediction, in the absence of MVR events, we still observed a distance-dependent increase in the event amplitude, and in the fast component of endocytosis, and this was the case for synchronous but not asynchronous release events (*Figure 5—figure supplement 2C*). This result indicates that the increase in the average synchronous event amplitude is not caused by the distance-dependent increase in the relative proportion of MVR.

A caveat in the above analysis comes from the fact that while smaller rings around the AZ center include events from most synapses, very distal rings include only events from the largest synapses, raising the possibility that part of the observed correlations could be an effect of the synapse size. To examine this possibility, we used instead the normalized distances to the AZ center. We obtained essentially the same results (*Figure 5—figure supplement 2A and B*) as those we observed with the actual distances, supporting that distance-dependent increase in synchronous event amplitude is not an effect of synapse size.

While amplitude variability and noisy nature of single events precluded the above analyses for individual events, we sought to confirm the distance-dependent differences between synchronous and asynchronous release at the level of individual boutons. For each bouton, we used the same concentric circle approach to average events detected within each distance band separately for synchronous and asynchronous events. The average event amplitude for each distance bin was then plotted as a function of distance to the AZ center separately for each bouton and fitted with a linear function (*Figure 5—figure supplement 2D*). We then quantified the percentage of boutons with a positive correlation between event amplitude and distance to the AZ center using a threshold of $R^2 > \pm 0.1$ to

mitigate the effects of noise. We observed that for synchronous events 65% of boutons showed a positive correlation, which was further increased to 68% when only UVR events were included in the analysis (*Figure 5J*). In line with the population analyses above, the presence of EGTA-AM effectively eliminated this phenomenon, bringing down the number of boutons with a positive correlation to chance (48%) (*Figure 5J*). Also, similarly to the population analysis, for asynchronous events, the number of boutons with a positive correlation was no higher than chance (52%) (*Figure 5J*). These results support the above population analyses at the level of individual boutons, and specifically the notion that synchronous and asynchronous release events differ in their coupling with the fast calcium-dependent component of endocytosis.

We also considered an additional contributing factor that variability in the timing of synchronous and asynchronous events, which determines the number of photons collected, could result in different localization precision in different locales across the AZ, thus contributing to observed differences in amplitude. To examine this possibility, we determined the error of event localization as a function of distance and found that it had little variation between synchronous events near the AZ center vs. periphery (*Figure 5—figure supplement 3A*), and between synchronous and asynchronous events (*Figure 1—figure supplement 1A*), and thus cannot account for the observed differences in event amplitude.

Finally, we considered the potential contribution from diffusion of vesicular components upon fusion, which could be different at the AZ center vs. periphery. The average spatial profile of synchronous release events had a similar width near the AZ center vs. periphery (*Figure 5—figure supplement 3B*), arguing against spatial differences in diffusion across the AZ. This is consistent with the observation that asynchronous events do not have any measurable spatial differences in amplitude or decay kinetics and thus in the contribution from diffusion in different locations of the AZ, and diffusion presumably affects various forms of release similarly. In addition, we considered that the membrane diffusion coefficient decreases approximately tenfold with the decrease in temperature from 37°C to room temperature (*Ries et al., 2009*), while we observed no significant changes in the kinetics of ultrafast endocytosis (*Figure 5—figure supplement 3F*) and only modest changes in the fast calcium-dependent component of endocytosis ($Q_{10} \sim 1.3$). Thus initial changes in the event spatial profile with decrease in temperature (if observed) can be attributed predominately to lateral diffusion. In contrast, we found that the event spatial profile was nearly the same at the two temperatures over the first three frames (150 ms) (*Figure 5—figure supplement 3C–F*), supporting the notion that diffusion does not play a major role in the observed differences in event amplitude or decay kinetics. Finally, photobleaching accounted only for ~3.3% of event decay during 1 s interval, and thus also cannot account for differences in event decay between the two forms of release.

Taken together, these observations suggest that while synchronous and asynchronous events generally utilize the same forms of retrieval, they differ in some aspects of exo-endocytosis coupling, particularly in the contribution of the fast calcium-dependent form of endocytosis.

## Spatiotemporal differences between synchronous and Sr²⁺-evoked asynchronous release events

In the above analyses, we defined asynchronous events based solely on temporal considerations. An alternative approach widely used previously to study asynchronous release relies on substitution of $Ca^{2+}$ for $Sr^{2+}$ (4 mM $Sr^{2+}$ application in 0.5 mM $Ca^{2+}$) (*Atluri and Regehr, 1998*; *Goda and Stevens, 1994*; *Rumpel and Behrends, 1999*; *Xu-Friedman and Regehr, 1999*). $Sr^{2+}$ enhances asynchronous release approximately three- to fourfold due to its slower clearance rate and differential affinity to Syt1 in comparison to calcium (*Atluri and Regehr, 1998*; *Goda and Stevens, 1994*; *Rumpel and Behrends, 1999*; *Xu-Friedman and Regehr, 1999*; *Dürst et al., 2022*). Indeed, we found that $Sr^{2+}$ application caused a significant reduction in synchronous release (p=0.0042, two-sample K-S test) and an increase in asynchronous release inside the AZ (p=0.028, two-sample K-S test; *Figure 6—figure supplement 1A–C*). This additional desynchronization in the presence of $Sr^{2+}$ provides a useful tool to enhance asynchronous release and examine to what extent the features we identified above are also present in this larger population of asynchronous events.

We first examined the spatial organization of $Sr^{2+}$-evoked asynchronous events. Specifically, we determined the proportion of asynchronous events released inside and outside the limits of the functionally defined AZ using the same approach and definitions as we did in *Figure 1D*. To avoid

confusion, we also keep the same terminology by referring to the events detected in the first frame as *Synchronous*, while the events detected in the second frame as *Asynchronous*. In the presence of $Sr^{2+}$, 76.5% of asynchronous events were located inside the AZ and 23.5% occurred ectopically, which is in the close agreement with the proportion we observed above in 'normal' control (2 mM $Ca^{2+}$) conditions (73.3/26.7%, *Figure 6A*). In fact, we did not find statistical differences between the spatial distributions of asynchronous events in the two conditions, neither for normalized distances (*Figure 6A*) nor for the absolute distances (*Figure 6C*) (p=0.79, K-S test). Also, similarly to our earlier observations, in the presence of $Sr^{2+}$ the subpopulation of asynchronous events inside the AZ was localized significantly closer to the AZ center than synchronous events in the same synapses (p=0.0067, K-S test; *Figure 6B*). The proportion of asynchronous events inside the AZ that utilize the same release sites as synchronous events was also similar to what we observed above in control conditions, with a modest increment (from ~25% to 31%, *Figure 6D*) and a small but significant reduction in distances to the release sites (p=0.0229, K-S test, *Figure 6D*). Taken together, these observations indicate a large degree of similarity in spatial organization between normal and $Sr^{2+}$-evoked asynchronous events, including the presence of ~25% of ectopic events, the spatial bias of the events inside the AZ toward the center, and the limited overlap with synchronous events at the same release sites. We note that the presence of $Sr^{2+}$ conveyed some changes to the spatial properties of 'synchronous' events (as defined by detection in first frame following stimulation), whose distribution was shifted in a complex way: it was similar to asynchronous events in normal conditions for shorter distances, but closer to synchronous events in normal conditions for longer distances. Thus 'synchronous' events display mixed synchronous/asynchronous-like spatial features in the presence of $Sr^{2+}$, suggesting that these events constitute a mixture of proper synchronous and $Sr^{2+}$-desynchronized events, with a variable degree of desynchronization that appears to correlate with the event distance to the AZ center.

Next, we examined the effects of $Sr^{2+}$ on the exo-endocytosis coupling. In contrast with the large differences in the decay kinetics between the two forms of release in control (normal) conditions, the average fluorescence traces for synchronous and asynchronous events were indistinguishable from each other in the presence of $Sr^{2+}$ (*Figure 6E*), including the time course and amplitude of the fast endocytosis (*Figure 6G*). In fact, in the presence of $Sr^{2+}$, the endocytosis kinetics of synchronous events became indistinguishable from asynchronous events in normal conditions (*Figure 6G*). Moreover, the distance-dependent increase in synchronous event amplitude and in the contribution of fast endocytosis was no longer observed in $Sr^{2+}$ (*Figure 6H*). This was the case both for the average traces across population (*Figure 6H*) and for single boutons (*Figure 6I*), and regardless of whether MVR events were included or not (*Figure 6I*). In contrast, the average traces of asynchronous events were similar in the presence of $Sr^{2+}$ to those in control conditions (*Figure 6F*). Also, similarly to control conditions, no distance dependence of decay kinetics was observed for asynchronous events in the presence of $Sr^{2+}$ (*Figure 6H*).

Taken together, these findings provide further support for the spatiotemporal features of asynchronous release we observed earlier by showing that asynchronous release events evoked by $Sr^{2+}$ and those we observed in normal conditions have very similar properties, including spatial organization, the presence of ectopic events, as well as release site utilization and decay kinetics. As expected from previous studies (*Atluri and Regehr, 1998*; *Goda and Stevens, 1994*; *Rumpel and Behrends, 1999*; *Xu-Friedman and Regehr, 1999*), the extensive desynchronization of release in the presence of $Sr^{2+}$ results in a greatly increased proportion of asynchronous events in the first frame following stimulation, thus causing a shift in the perceived properties of 'synchronous' events (as defined in our study) towards the properties of asynchronous events.

## Discussion

Asynchronous release is widely recognized to play important roles in synaptic transmission, yet its organization at the AZ and interrelationship with synchronous release have remained debatable. Here we compared nanoscale localization of synchronous and asynchronous release events within the same presynaptic boutons to identify two diverse populations of asynchronous events with distinctive organization and spatiotemporal dynamics.

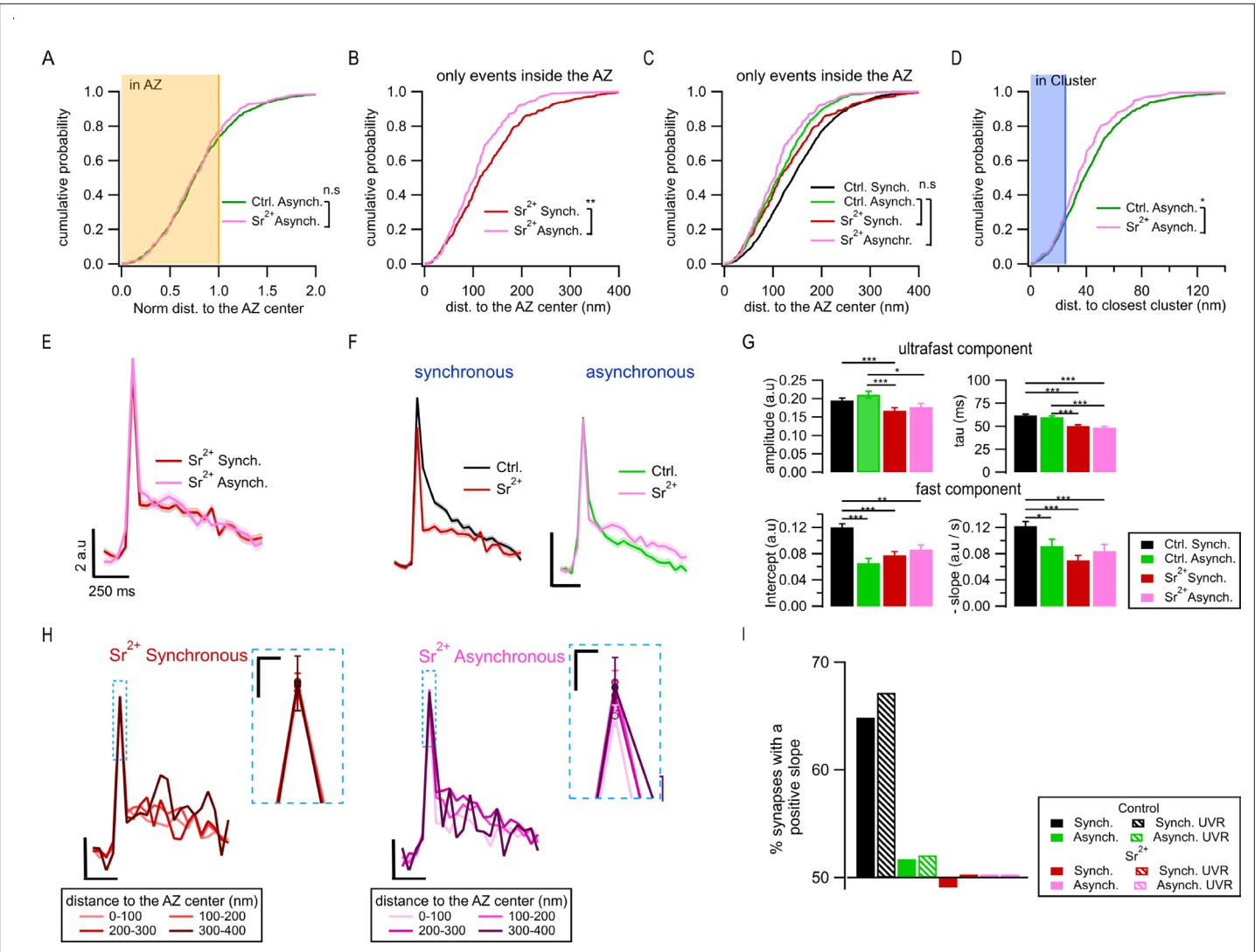

**Figure 6.** Spatiotemporal properties of Sr$^{2+}$-evoked asynchronous release events. (**A**) Cumulative distributions of the normalized distances to the active zone (AZ) center for asynchronous events recorded under control conditions (green, N = 1089) and those recorded in 4 mM Sr$^{2+}$ (see 'Materials and methods'; pink, N = 232; n.s., p=0.79, K-S test). The shaded area represents the fraction of events localized inside the AZ (normalized distance to the center ≤ 1). (**B**) Cumulative distributions of the distances to the AZ center for synchronous events (red line, N = 392) and asynchronous events localized inside the AZ (pink, N = 177), both recorded in Sr$^{2+}$ (**p=3.3e-4, K-S test). (**C**) Same data as in (**B**), plotted together with their equivalents in control conditions: synchronous (black) and asynchronous (green) (same data as in **Figure 1G**). Only the nonsignificant comparisons are shown (all the other comparisons are given in **Supplementary file 1**). (**D**) Cumulative distributions of the distances to the closest cluster/release site for asynchronous events in control conditions (green, N = 799) and in Sr$^{2+}$ (pink, N = 177). (*p=0.023, K-S test). The shaded area highlights the proportion of events that were considered to be in a cluster. (**E**) Mean pHluorin signals of synchronous (red, N = 1413) and asynchronous (pink, N = 917) events recorded in Sr$^{2+}$ with SEM displayed as a shading. (**F**) Comparison of the mean normalized pHluorin signals (Min-max normalization of each individual trace: trace – min(trace)/max(trace) – min(trace)) of synchronous (left) and asynchronous (right) events under control and Sr$^{2+}$ conditions. Calibration bar: 0.1 a.u., 250 ms. (**G**) Quantification of the ultrafast component (upper bar graphs: amplitude and tau) and fast component (bottom bar graphs: intercept and slope) of the pHluorin signal decay for synchronous and asynchronous events in control and Sr$^{2+}$ conditions, as indicated (***p<0.0001, Kruskal–Wallis test, all statistical comparisons are given in **Supplementary file 1**). (**H**) Spatial analysis of the pHluorin signal in Sr$^{2+}$ conditions using the same approach and bin size as in **Figure 5E**. Mean pHluorin signals per spatial bin are shown for synchronous (red) and asynchronous (pink) events in the presence of Sr$^{2+}$. Insets show zoom-in of the peaks. Calibration bars: 2 a.u., 250 ms. Inset: 1 a.u., 25 ms. (**I**) Proportion of synapses that show a positive correlation between the amplitude of the pHluorin signal and the distance to the AZ center (see 'Materials and methods') for the two forms of release in control and Sr$^{2+}$ conditions. Bars labeled as univesicular (UVR) correspond to the analyses in which we removed multivesicular (MVR) events (i.e. UVR-only conditions). The values are plotted relative to 50%, which represents the probability of observing a positive or negative correlation by chance. Sr$^{2+}$ (red and pink) eliminated the positive correlation between the amplitude of synchronous events and distance to the AZ center.

The online version of this article includes the following figure supplement(s) for figure 6:

**Figure supplement 1.** Application of Sr$^{2+}$ causes desynchronization of release.

## Two populations of asynchronous release events

Our results suggest that asynchronous release is not a homogeneous population simply distributed in time, but it contains at least two distinct subpopulations: one occurring inside the AZ and another occurring ectopically. This notion is supported by the opposing preferential localization of the two subpopulations, which cannot be explained by a single homogeneous population of events artificially divided by a specific definition of the AZ boundary in our measurements. Moreover, the ectopic and in-AZ events differ markedly in temporal properties since the ectopic events occur with significantly larger time intervals from synchronous events than the asynchronous events inside the AZ, thus further supporting the presence of two distinct subpopulations. The strong spatial bias toward the AZ center that we observed for the asynchronous release events inside the AZ is in a good agreement with the previous study using flash-freeze EM (*Kusick et al., 2020*; *Li et al., 2021*), while the subpopulation of ectopic events may have been overlooked in that study because a single time point used was likely insufficient to capture the much less frequent ectopic events. However, our results are also in agreement with the seemingly incompatible live imaging study of asynchronous events using SF-iGluSnFR (*Mendonça et al., 2022*) because we also observed that the combined population of all asynchronous events occurs farther away from the AZ center than synchronous events and thus occupy a larger release area, as measurements with SF-iGluSnFR reported (*Mendonça et al., 2022*). Our observation that there are two spatially distinct subpopulations of asynchronous release events thus reconciles the previous studies suggesting diametrically opposing organization of the asynchronous release at the AZ.

Although, to the best of our knowledge, ectopic asynchronous release has not been previously reported, there is extensive evidence for ectopic synchronous release occurring outside of the ultrastructurally defined AZ in several types of synapses, including cerebellar synapses (*Matsui and Jahr, 2003*; *Matsui and Jahr, 2004*; *Matsui et al., 2005*), ribbon-type synapses (*Lenzi et al., 2002*; *Zenisek et al., 2000*), and several glutamatergic and cholinergic sensory synapses (*Balakrishnan et al., 2014*; *Balakrishnan et al., 2011*; *Coggan et al., 2005*; *Fu and Sretavan, 2012*; *Thyssen et al., 2010*). Synchronous ectopic release was found to rely on a different proportions of voltage-gated $Ca^{2+}$ channel (VGCC) subtypes than release within the AZ (*Matsui and Jahr, 2004*) and to exhibit different release probability and short-term plasticity (*Dobson and Bellamy, 2015*). Despite the differences in spatiotemporal organization, we found that ectopic asynchronous release events also share many similarities with the release events within the AZ, including event waveform, suggesting some similarity in the release machinery. Indeed, at the level of molecular organization, release sites are associated with the nanoclusters of SNARE proteins and docking machinery (*Gramlich and Klyachko, 2019*) and extensive evidence indicates that these nanoclusters are not limited to the AZ, but are also commonly present at extrasynaptic locations including the axons (*Garcia et al., 1995*; *Sesack and Snyder, 1995*; *Hagiwara et al., 2005*). Moreover, STED measurements showed that clusters of a SNARE protein Syntaxin-1A are found all over the synaptic membrane at the neuromuscular junctions, well beyond the boundaries of the AZ (*Ribrault et al., 2011*; *Ullrich et al., 2015*). Syntaxin-1A clusters are more abundant, stable, and larger in size inside the AZ in comparison to ectopic clusters (*Ullrich et al., 2015*). Syntaxin-1A co-clusters with the L-type VGCCs (*Sajman et al., 2017*), suggesting that the smaller outside clusters may serve as sites of ectopic release (*Ullrich et al., 2015*). Future studies combining functional measurements of individual vesicle release events will nanostructural measurements of synaptic docking and release machinery will be needed to determine the precise molecular determinants of ectopic release sites.

## Release site utilization by synchronous and asynchronous release

One of the major unresolved questions regarding the relative spatial arrangement of synchronous and asynchronous release is whether and to what extent the two forms of release have a spatial overlap at a nanoscale and utilize the same or distinct release sites. Flash-freeze EM studies suggested entirely distinct organization of the two forms of release with a preferential trans-synaptic alignment of asynchronous release events with a cluster of NMDA receptors near the AZ center, while synchronous events align with more peripherally located clusters of AMPA receptors (*Kusick et al., 2020*; *Li et al., 2021*). This preferential trans-synaptic alignment with the spatially distinct clusters of postsynaptic receptors suggests that different release sites are involved in supporting the two forms of release. In contrast, live imaging studies with SF-iGluSnFR observed ~50% overlap of the two forms of release at

the same release sites, pointing to a large degree of overlap, and arguing against preferential alignment with two spatially segregated types of receptors (*Mendonça et al., 2022*). Our results again are compatible with and largely reconcile the two previous studies by showing that the two forms of release can overlap at the same release sites, but only in ~15–20% of the cases, at least within our limited observation time window.

There are two main constraints that affect how the number of release sites and spatial overlap of different forms of release is determined in ours and previous studies. First, the duration of observation likely contributes to the observed differences between the studies. By design, the flash-freeze EM studies were limited to observing a single event per synaptic bouton (*Kusick et al., 2020*; *Li et al., 2021*), and thus the overlap of synchronous and asynchronous release events could only be inferred indirectly from distributions across different boutons. A live imaging study with the SF-i-GluSnFR compared the two populations within the same bouton, but had another caveat in having an extensive period of observation lasting several minutes (*Mendonça et al., 2022*). We previously found that analyses of release site utilization in our culture conditions, which are similar to the above study (*Mendonça et al., 2022*), must be limited to ~120–150 s by the random natural displacement of synapses in culture, which has a rate of ~20–25 nm/min (*Maschi and Klyachko, 2017*), and cannot be accounted for as it varies from one bouton to another. As a result, longer recordings cause a progressively increasing blur in the definition of release sites based on event clustering, effectively broadening the release site dimensions and thus overestimating the spatial overlap between the two forms of release. Second, the diameter of the cluster used in hierarchical clustering algorithms to define the release sites is also critical in determining the spatial degree of overlap between different forms of release. The cluster diameter used in our studies is derived independently from any biological considerations by calculating the inconsistency coefficient, as required by the algorithm (*Wang et al., 2016*), giving the value for the natural cluster diameter in our data of ~50 nm (*Maschi and Klyachko, 2017*). This value is also supported by the estimates for the number of release sites per bouton we obtained using 50 nm clusters, which is in a good agreement with the estimates produced by an entirely different approach using quantal analysis of glutamate transients (*Sakamoto et al., 2018*). The localization precision reported for SF-iGluSnFR of ~75 nm (*Mendonça et al., 2022*) does not permit event clustering with effective diameter smaller than 150 nm. Given the average AZ dimensions of only ~250 nm (*Ackermann et al., 2015*), the cluster diameter of 150 nm would encompass a large proportion of the AZ, thus likely overestimating the degree of overlap between the two forms of release. Thus within the constraints of the limited observation period, our results reconcile the two previous studies, suggesting that synchronous and asynchronous events can indeed overlap at the same release sites, but to a large extent are spatially segregated.

## Exo-endocytosis coupling of synchronous and asynchronous release

A surprising observation that further distinguishes synchronous and asynchronous release events is the difference in the average event amplitude. This difference arises from a systematic increase in the synchronous event amplitude with the distance from the AZ center, while the amplitude of asynchronous events remained uniform throughout the AZ in the same boutons. Our analyses indicate that this is not an artifact, which can simply arise from a different duration of acquisition for synchronous vs. asynchronous events, because this phenomenon is not observed for synchronous and asynchronous events near the AZ center. Moreover, this distance-dependent phenomenon was eliminated by buffering intra-terminal calcium with EGTA and was no longer observed when $Ca^{2+}$ was substituted for $Sr^{2+}$, suggesting that it is caused by a calcium-dependent mechanism that differs between synchronous and asynchronous forms of release. To understand the possible causes of this phenomenon, we considered that the average amplitude of single release events is determined by several factors: the amount of fluorophore per vesicle, the relative contribution of MVR, the kinetics of lateral diffusion of vesicle components in the plasma membrane, the timing of fusion events, and the efficiency of endocytosis. We can exclude the first factor for several reasons: it is highly unlikely that vesicles within the same bouton have a systematically different amount of pHluorin that scales with distance to the AZ center, and in a calcium-dependent manner. Furthermore, it is also highly unlikely that vesicles undergoing synchronous release would have systematically larger pHluorin load than those released asynchronously. We also considered the possibility that the differential amount of MVR among synchronous and asynchronous events, and/or a spatial gradient of MVR from AZ

center to periphery could contribute to observed differences in the event amplitude between the two forms of release. However, we found that excluding MVR events made the spatial gradient and the differences between the two forms of release larger, rather than eliminated them. In consideration of the third possibility, we found that the contribution from lateral diffusion of vesicular components to apparent event decay does not differ between the AZ center and its periphery. We also note that the differences in lateral diffusion between the AZ center and periphery are highly unlikely to explain this phenomenon because a less dense membrane environment at the periphery of the AZ (*Gramlich and Klyachko, 2019*; *Südhof, 2012*) would be expected to allow for more efficient lateral diffusion and decay of the fluorescence signal, while the opposite is observed. Moreover, the two forms of release have different amplitudes at the same distance from the AZ center, where the contribution from diffusion is presumably similar. Finally, differences in the timing of vesicle fusion, which determines the amount of photons collected, could contribute to the observed differences in amplitude. In addition to a variable delay of asynchronous events comparing to synchronous ones, difference in the event timing could arise, for example, from different distance to the calcium source at different locales of the AZ. If this is the case, the increase in synchronous event amplitude with distance from the AZ center would imply systematically earlier vesicle fusion and closer distance to the calcium source at the AZ periphery. While we cannot directly exclude this possibility, we previously observed that release sites near the AZ center have several fold higher release probability and are affected stronger by EGTA than the release sites near the AZ periphery (*Maschi and Klyachko, 2017*). Central release sites are thus presumably closer to the calcium sources, which is opposite in direction of changes from the distance-dependent effect we observed here. Furthermore, we observed no measurable differences in amplitude between synchronous and asynchronous events near the AZ center, suggesting that the variability in event timing, while present, is not the principal determinant of the event amplitude in our measurements. Finally, the error of event localization, which reflects the number of photons collected, had little variation between synchronous vs. asynchronous events and between events at the AZ center vs. the periphery, further arguing against the critical role of event timing in this phenomenon. Thus the most likely remaining explanation is differential exo-endocytosis coupling or endocytosis kinetics between the two forms of release. This is also supported by the observed differences in the decay kinetics of synchronous and asynchronous events, and by the observation that the difference in the average event amplitude between the two forms of release was eliminated by acute inhibition of dynamin.

Previous analyses of single-vesicle release events have demonstrated that synchronous release can be coupled with several kinetically and mechanistically distinct forms of endocytosis (*Zhang et al., 2013*). Whether different forms of release, that is, synchronous and asynchronous, are coupled with the same or different forms of endocytosis has not been explored. Our analyses showed that both synchronous and asynchronous events decay with two clearly distinguishable kinetic components: one with a time constant ~60 ms and calcium-independent, and another one with a timescale of about a second and calcium-dependent. These two decay components closely correspond in both kinetics and calcium sensitivity to the previously identified ultrafast endocytosis and fast endocytosis, respectively (*Chanaday and Kavalali, 2018*; *Watanabe et al., 2013*; *Zhang et al., 2013*). Our results suggest that while the two forms of release generally utilize the same forms of retrieval, there are differences in some aspects of their exo-endocytosis coupling. The role of endocytosis in the observed differences is also consistent with the effects of EGTA, whose action on the fast component of endocytosis parallels its action on the synchronous event amplitude. This observation is in line with recent studies suggesting that calcium influx may activate calmodulin and synaptotagmin to initiate endocytosis and control endocytosis kinetics (*Wu et al., 2009*). Indeed, increasing calcium influx during APs accelerates endocytosis at hippocampal synapses (*Balaji et al., 2008*; *Sankaranarayanan and Ryan, 2001*), while buffering intra-terminal calcium elevation with EGTA slows down fast endocytosis at bipolar nerve terminals and in the calyx of Held synapses (*Neves et al., 2001*; *Wu et al., 2005*). Another possible mechanism is differential distribution or mobility of SNARE proteins across the AZ since the SNARE proteins have been suggested to control the exo-endocytosis coupling (*Wu et al., 2014*; *Zhang et al., 2013*), and a large variability in cluster size and mobility of SNARE proteins has been found in different locales of the AZ (*Ribrault et al., 2011*; *Ullrich et al., 2015*). Future studies will be needed to define the complex mechanisms of exo-/endocytosis coupling and their interrelationship with the two forms of release. We note that while we did not observe differences in the ultrafast

component of endocytosis between the two forms of release, the limited temporal resolution of our recordings and/or temporal limitations imposed by the speed of reacidification and quenching of the fluorescence probe may have prevented an observable difference in kinetics.

Nevertheless, our results uncover a previously unrecognized complexity in spatial and temporal organization of asynchronous release in central synapses, whose distinctive features and dynamic balance with the synchronous release may play important roles in synaptic computations.

# Materials and methods

**Key resources table**

| Reagent type (species) or resource | Designation | Source or reference | Identifiers | Additional information |
|---|---|---|---|---|
| Transfected construct (synthetic) | pFU-vGluT1-pHGFP-W | Viral Vectors Core at Washington University | n/a | |
| Transfected construct (synthetic) | pAAV.hSynapsin.SF-iGluSnFR.A184S | Addgene | 106174-AAV1 | |
| Biological sample (rat, Long-Evans) | Hippocampus of rat pups | Charles River | 006 | |
| Chemical compound, drug | APV | Sigma-Aldrich | 79055-68-8 | |
| Chemical compound, drug | CNQX disodium salt hydrate | Sigma-Aldrich | 115066-14-3 | |
| Chemical compound, drug | Minimum Essential Media (MEM) – no phenol red | Gibco | 51200038 | |
| Chemical compound, drug | Characterized fetal bovine serum | Gibco | 10437028 | |
| Chemical compound, drug | Penicillin-streptomycin (5000 U/ml) | Gibco | 15070063 | |
| Chemical compound, drug | N-2 Supplement (100×) | Gibco | 17502048 | |
| Chemical compound, drug | Neurobasal-A Medium | Gibco | 12349015 | |
| Chemical compound, drug | B-27 supplement (50×), serum free | Gibco | 17504044 | |
| Chemical compound, drug | GlutaMAX Supplement | Gibco | 35050061 | |
| Chemical compound, drug | PDL(poly-D-lysine) | BD Biosciences | 40210 | |
| Chemical compound, drug | Trypsin-EDTA (0.05%), phenol red | Gibco | 25300054 | |
| Software, algorithm | MATLAB | MathWorks | RRID:SCR_001622 | |
| Software, algorithm | u-track 2.0 | https://www.utsouthwestern.edu/labs/danuser/software/#utrack_anc; (*Aguet et al., 2013*; *Jaqaman et al., 2008*) | | |

## Neuronal cell culture

Neuronal cultures were produced from the hippocampus of rat pups of mixed gender as previously described (*Murthy and Stevens, 1999*; *Peng et al., 2012*). Briefly, hippocampi were dissected from E16-17 pups, dissociated by papain digestion, and plated on glass coverslips containing an astrocyte monolayer. Neurons were cultured in Neurobasal media supplemented with B-27 supplement.

All animal procedures conformed to the guidelines approved by the Washington University Animal Studies Committee.

## Viral infection

VGluT1-pHluorin was generously provided by Drs. Robert Edwards and Susan Voglmaier (UCSF) (*Voglmaier et al., 2006*; *Silm et al., 2019*). SF-iGluSnFR(A184) was kindly made available by Dr. Loren Looger (Addgene viral prep #106174-AAV1) (*Marvin et al., 2018*). Viral vectors were generated by the Viral Vectors Core at Washington University or obtained from Addgene. Hippocampal neuronal cultures were infected at DIV3 as previously described (*Maschi and Klyachko, 2017*) and experiments were performed at DIV16–19.

## Fluorescence microscopy

Fluorescence was excited with a 475 nm LED (Olympus) and monitored using an inverted microscope (IX83, Olympus) equipped with a ×150 1.45 NA oil-immersion objective. A continuous acquisition was performed at 50 ms/frame for 120 s using a cooled EMCCD camera (iXon life 888, ANDOR). The Z-drift compensation system (IX3-ZDC, Olympus) was used to ensure a constant position of the focal plane during imaging. All experiments were conducted at 37°C within a whole-microscope incubator chamber (TOKAI HIT). Field simulation was performed at 1 Hz precisely synchronized with the beginning of an acquisition frame using a pair of platinum electrodes and controlled by the software via Master-9 stimulus generator (A.M.P.I.). Samples were perfused with bath solution containing 125 mM NaCl, 2.5 mM KCl, 2 mM $CaCl_2$, 1 mM $MgCl_2$, 10 mM HEPES, 15 mM glucose, 50 µM APV, 10 µM CNQX, adjusted to pH 7.4.

## Near-TIRF imaging

Fluorescence was excited with a 488 laser (Cell CMR-LAS-488, Olympus) and monitored using an inverted TIRF-equipped microscope (IX83, Olympus) under a ×150/1.45NA objective. Near-TIRF was achieved by adjusting the incident angle to 63.7°, which is near the critical angle of 63.63°. Images were acquired every 50 ms (with an exposure time of 49.38 ms) using a cooled EMCCD camera (iXon life 888, ANDOR). In experiments in *Figure 1H, I*, acquisition was performed at a rate of 25 ms/frame. Other experimental details, including solutions, field stimulation, etc., were the same as above.

## Quantification and statistical analyses

### Event detection and localization

The detection and subpixel localization of individual vesicle release events were performed as previously described (*Maschi and Klyachko, 2017*; *Maschi and Klyachko, 2020*) using MATLAB and the uTrack software package, which was kindly made available by Dr Gaudenz Danuser's lab (*Aguet et al., 2013*; *Jaqaman et al., 2008*). Error of event localization (localization precision, dx) was determined from least-squares Gaussian fits of individual events using in-built functions in uTrack software (*Aguet et al., 2013*; *Jaqaman et al., 2008*), as previously described (*Maschi and Klyachko, 2017*; *Maschi and Klyachko, 2020*).

### Identification of asynchronous release events

Asynchronous events were defined using two approaches. First, we used temporal considerations to distinguish asynchronous from synchronous release. Release events detected in the second frame following an AP were considered to represent asynchronous events in our measurements because they had a delay of at least 50 ms following an AP. Since only the events that occur within the first ~10–20 ms of the frame produce sufficient amount of photons to be above the detection limit, the population of events detected in the first frame was effectively limited to a large extent to synchronous events. Because the fluorescence signal from synchronous events can occasionally persist from the first into the second frame, to avoid misidentification of such events as asynchronous, events detected in the second frame were included in analysis only if there was no release event detected in the preceding (first) frame in the same synapse. For measurements shown in *Figure 6*, an alternative approach was used by substituting $Ca^{2+}$ for $Sr^{2+}$ (4 mM $Sr^{2+}$ application in 0.5 mM $Ca^{2+}$), an approach that has been widely used previously to enhance and study asynchronous release (*Atluri and Regehr, 1998*; *Goda and Stevens, 1994*; *Rumpel and Behrends, 1999*; *Xu-Friedman and Regehr, 1999*).

## Identification of multivesicular release events

For each individual synapse, all detected event amplitudes at the frame of localization were baseline-subtracted, pooled together, and tested for normality (the baseline, F0, was defined as the average signal during the five frames before detection). In the absence of MVR events, these amplitudes are expected to follow a single Gaussian distribution whose average would correspond to the mean quantal value per synapse. Consequently, we tested if the amplitudes follow a normal distribution using the Anderson–Darling test. If normality was proven, we divided each amplitude value by the average; if not, we removed the highest amplitude each time and tested again for normality until reached. The average of the normally distributed subset was used to normalize all the events. The events whose normalized amplitude was larger than 2 were considered to be MVR, corresponding to ~4% of synchronous and ~1.4% of asynchronous events.

## Definition of AZ dimensions and center

The AZ size was approximated based on the convex hull encompassing all synchronous release events in a given bouton. This definition is in a close agreement with the ultrastructural measurements of AZ dimensions (*Maschi and Klyachko, 2017*). AZ center was defined as the mean position of all synchronous release events in a given bouton as described (*Maschi and Klyachko, 2017*).

## Normalization of the distance measurements for release events

Distance normalization was performed using the line connecting the AZ center and the event's location, such that the distance from the AZ center to the event was normalized to the distance from the AZ center to the border of the AZ along the same line (*Figure 1D*). A 25 nm wide rim was added to the AZ border to account for the uncertainty in the localization of the events that define the AZ area. Events with normalized distance to the AZ center ≤1 were classified as being inside the AZ; events with normalized distance >1 were classified as being outside the AZ.

## Definition of release sites

Release sites were defined using hierarchical clustering algorithm with a clustering diameter of 50 nm using built-in functions in MATLAB as described (*Maschi and Klyachko, 2017*; *Maschi et al., 2018*). We have previously shown that the observed clusters do not arise from random distribution of release events, but rather represent a set of defined and repeatedly reused release sites within the AZs.

## Simulations of random spatial distribution of release events

To compare the location of asynchronous events with a random distribution, the locations of the asynchronous events in each bouton were simulated by randomizing them over the original AZ defined by the synchronous events. Using the center of the AZ as pole to define the new polar coordinates, the radial coordinates for each simulated asynchronous event were randomly selected from uniformly distributed numbers from 0 to $2\pi$. The radial coordinates were chosen randomly following the same probability distribution of all asynchronous events.

## Quantification of the two components of endocytosis

A circular ROI with a radius of 5 pixels centered at the localization pixel was used for each event. Given point spread function (PSF) sigma (1.38 pixels) of our system, this size of the ROI is sufficient to capture the entire signal of individual release events. Based on this ROI, the pHluorin signal amplitude of each event was determined for 10 frames preceding the event detection and all subsequent frames until the following stimulus. To quantify the two temporal components of the pHluorin signal decay, we first fit a line to the final portion of the trace starting from the fifth point after the detection. This fit was extrapolated to the beginning of the trace and its parameters (slope and intercept) were taken to correspond to the fast component of endocytosis. The linear component was subtracted from the trace and the remaining signal was fitted using a single exponential function. The amplitude and tau of the exponential fit were taken to represent parameters of the ultrafast endocytosis. This approach was more robust than a double exponential fit given the small amplitude and low signal-to-noise ratio of single event traces. In addition, it allowed us to describe the two components of decay independently.

## Spectral analysis of event time course

For each synchronous and asynchronous event, the pHluorin signal time course was determined as described above and the discrete Fourier transform was computed using MATLAB's Fast Fourier Transform algorithm. The first 900 ms of the signal, starting at the moment of detection, were analyzed to ensure the same number of points for synchronous and asynchronous events. Then we computed the single-sided spectrum based on the two-sided spectrum and the even-valued signal length.

## Analysis of the distance dependence of the average event amplitude

Pearson's Linear Correlation algorithm was used to determine the correlation coefficient and significance of correlation between the event amplitude and its distance to the AZ center. The results were the same regardless of the number of bins selected: the correlation coefficient was always highly positive with a $p < 0.05$ for synchronous release, but near 0 and with $p > 0.05$ for asynchronous release. This was also confirmed with an automatically selected binning by the Freedman–Diaconis rule.

## Analysis of the distance dependence of event amplitude at individual synapses

For every release event at each individual synapse, the magnitude of the pHluorin signal at the frame of detection was measured and background subtracted, which was defined as the average signal during the five frames before detection. The relation between the events' amplitudes and their distance to the AZ center was binned (with 80 nm bins) and fitted by a linear function. Synapses with less than three bins were excluded. This analysis was carried out in a subset of 100 synapses with the largest number of events to make the fit more reliable. Finally, the synapses with fits those $R^2 < \pm 0.1$ were excluded, as not robustly representing either positive or negative slopes. Finally, the proportion of synapses that had a positive slope was determined in each condition.

## Data inclusion and exclusion criteria

To minimize contributions from the AZs that are strongly tilted relative to the imaging plane and therefore are only partially in focus, a minimum of five detected synchronous release events per bouton was required for all analyses, except where stated otherwise. To ensure adequate sampling of release sites during our limited observation period, a minimum of 10 release events per bouton was required for all spatial analyses, except where stated otherwise.

## Statistical analysis

Statistical analyses were performed in MATLAB. Statistical significance was determined using a two-sample two-tailed *t*-test, Tukey–Kramer ANOVA, Kolmogorov–Smirnov (K-S), or Kruskal–Wallis tests where appropriate. Statistical tests used to measure significance, the corresponding significance level (p-value), and the values of n are provided in *Supplementary file 1*. Data are reported as mean ± SEM, and $p < 0.05$ was considered statistically significant.

## Acknowledgements

This work was supported in part by the R35 grant to VAK from NINDS and postdoctoral fellowship from the McDonnell Center for Molecular Neuroscience at Washington University to GM.

## Additional information

### Funding

| Funder | Grant reference number | Author |
| --- | --- | --- |
| National Institute of Neurological Disorders and Stroke | R35NS111596 | Vitaly A Klyachko |

| Funder | Grant reference number | Author |
| --- | --- | --- |
| McDonnell Center for Cellular and Molecular Neurobiology, Washington University in St. Louis | | Gerardo Malagon |

The funders had no role in study design, data collection and interpretation, or the decision to submit the work for publication.

## Author contributions

Gerardo Malagon, Software, Formal analysis, Validation, Investigation, Visualization, Methodology, Writing – original draft, Writing – review and editing, Performed all epifluorescence microscopy experiments (Figures 1A-G, Figures 3-6), wrote the first draft of the Results and Methods; Jongyun Myeong, Software, Formal analysis, Validation, Investigation, Visualization, Methodology, Writing – original draft, Writing – review and editing, Performed all near-TIRF based experiments (Figures 1H,I and Figure 2); Vitaly A Klyachko, Conceptualization, Resources, Software, Supervision, Funding acquisition, Methodology, Writing – original draft, Project administration, Writing – review and editing

## Author ORCIDs

Jongyun Myeong ![ORCID] http://orcid.org/0000-0002-7900-0884
Vitaly A Klyachko ![ORCID] http://orcid.org/0000-0003-3449-243X

## Decision letter and Author response

Decision letter https://doi.org/10.7554/eLife.84041.sa1
Author response https://doi.org/10.7554/eLife.84041.sa2

# Additional files

## Supplementary files
- Supplementary file 1. Table containing the statistical information for all figures.
- MDAR checklist

## Data availability

This paper does not report standardized data types. The data generated and analyzed in this study are included in the manuscript, figures and supplemental information and can be found at https://doi.org/10.5061/dryad.fqz612jww. This study did not generate new or unique reagents or other materials.All original Matlab code and analyses routines developed in this work are available at https://github.com/gmalagonv/2-forms-of-asynchronous-release (copy archived at *Malagon, 2022*).

The following dataset was generated:

| Author(s) | Year | Dataset title | Dataset URL | Database and Identifier |
| --- | --- | --- | --- | --- |
| Klyachko V, Malagon G, Meyong J | 2023 | Data from: Two forms of asynchronous release with distinctive spatiotemporal dynamics in central synapses | https://dx.doi.org/10.5061/dryad.fqz612jww | Dryad Digital Repository, 10.5061/dryad.fqz612jww |

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
