## [Editor Report]

Spatial organization of distinct forms of release within individual synapses is a key open question in synaptic transmission. The authors provide novel insight into this question using state-of-the-art approaches to visualize asynchronous release events and their location. Their data pushes the boundaries of this form of analysis and suggests an intricate functional nano-organization within single active zones and their periphery. These findings present a novel perspective for synaptic signaling occurring at the single release site level.

---

## [Decision Letter]

**Decision letter after peer review:**

Thank you for submitting your article "Two forms of asynchronous release with distinctive spatiotemporal dynamics in central synapses" for consideration by *eLife*. Your article has been reviewed by 3 peer reviewers, one of whom is a member of our Board of Reviewing Editors, and the evaluation has been overseen by Lu Chen as the Senior Editor. The reviewers have opted to remain anonymous.

Essential revisions:

While the reviewers are excited about this study, they have also raised concerns. In essence, the reviewers acknowledge that the authors needed to push the limits of resolution (in terms of identifying optimum imaging duty cycles and photon counts) to address spatial and temporal aspects of single fusion events within a release site. However, these "optimum" settings also raise two key issues that require further attention by the authors.

1) Spatial and temporal dissociation of synchronous and asynchronous fusion events: As indicated by the reviews, the existing analysis requires further justification and validation. In particular, further metrics on the reliability and accuracy of event detection in terms of spatial localization and temporal dynamics of events are needed. How accurate are the distinctions between synchronous and asynchronous events? What are the estimates for false detection, or overlap?

2) The reviewers were less enthusiastic about the findings on differential coupling of vesicle fusion and retrieval kinetics due to temporal limitation of the super-resolution method. The issues raised above on accurate event detection present severe confounds on analysis of vesicle retrieval as this measure is complicated by re-acidification kinetics and lateral diffusion of probes. Here, the authors need to provide further analysis and/or data validating their conclusions, in particular clearly identifying sources of error and stating the error margins on their results.

*Reviewer #1 (Recommendations for the authors):*

1. A quantification of the individual synaptic preferences for different types of release would be informative – did some synapses primarily release vesicles via synchronous release, active zone asynchronous, or ectopic asynchronous release?

2. Since the authors already have the data, it would be interesting to quantify a nanocolumn phenotype at each active zone – specifically the number of active zones that display > 1 clusters of release sites such as in Tang et al. 2016.

3. What percentage of fluorescence decay is ultra-fast vs. fast – does the increased intercept for synchronous release mean a higher percentage of individual events go through calcium dependent endocytosis? Quantify the percentage for each type of release.

4. Synchronous and asynchronous events having the same kinetics for ultra-fast endocytosis could be at limitation of the speed of reacidification and quenching of the fluorescence probe, preventing an observable difference in kinetics. Please discuss this in the discussion.

5. Quantification of the amplitude of synchronous and asynchronous release at same distances from AZ center would be informative – I am assuming based on example traces that at the smallest distance synchronous and asynchronous are the same.

6. The addition of more methodology details interspersed in the Results section as opposed to just citing previous work from the lab would help in ease of reading (it can be brief). i.e. – how were average traces made (averaged for the same bouton, one field of view, etc.), how were MVR events excluded from analysis?

7. The addition of cartoons/graphics to make figures more legible would be helpful – i.e. in Figure 1 expanding Ai and Aii to include acquisition rate info, a stimulation protocol illustration, timeline of observation period, especially since a study design caveat is if all release sites were observed during the observation period

*Reviewer #2 (Recommendations for the authors):*

1) The authors looked to investigate coupling of exo/endocytosis and clustered events in 100nm rings and observed a "gradual and significant increase in the amplitude of synchronous events from the center of the AZ towards periphery". It is unclear what the statistics are to differentiate this small difference. It seems possible given the lower temporal measurements of 50ms time frames makes these minor changes liable to differences in time of timing of vesicle fusion in relation to calcium sources. Is it possible that distance to a calcium source might change the timing of the synchronous phase of vesicle fusion?

2) In relation to the above comments, given the imaging window compared to acquisition, would this be changing observed peaks of fluorescence, that could translate into changes in magnitude as well as fitting for endocytosis?

3) It seems that a limited number of hypothesis are tested and that the limited speed of imaging might be causing the differences observed. It is interesting that this difference is not seen in other conditions, asynchronous, Sr2+ and EGTA, but these are all altering the time locking of calcium-evoked fusion as well. Given the rather small nature of the difference that seems close to the St. Dev. of the noise as shown in Figure 3A and slow nature of imaging required by the probes, this part of the manuscript is less convincing.

*Reviewer #3 (Recommendations for the authors):*

The work is a careful analysis of individual vesicle fusion events visualized following stimulation. The authors use detailed analysis to parse the spatial and temporal features of fusion events within 50 ms of an AP (synchronous) and at later time points (asynchronous). While I think the findings are interesting, there are a few concerns that I feel the authors should address.

1. I am concerned about the possible confounds between the timing of fusion events, the number of photons collected and the spatial resolution. In particular, the spatial resolution is known to be a function of the SNR for any particular event and thus dimmer events will result in poorer spatial precision in the measurements. The authors state that the stimulus is precisely timed to the beginning of a frame and since most synchronous events will occur soon after, they should integrate the signal over the entire frame. The asynchronous events may occur at any point during the frame and will thus be expected to have a dimmer peak intensity (figure 3F seems to confirm this difference) and presumably poorer spatial resolution. The relationship between distance to the center and the intensity in figure 3F seems to partially argue against this, but doesn't fully address whether some of the apparent differences in localization of asynchronous effects might result from differences in precision of localization (for example, the localization to release sites and the finding of ectopic events).

2. The analysis of endocytosis is performed by looking at the kinetic features of fluorescence decay following exocytosis. Other groups have found that approximately 20% of synaptic vesicle proteins escape the boutons by diffusion. To what extent is the endocytosis signal diffusion out of the ROI? Is there an estimate to the expected loss due to photobleaching?

3. The analysis on page 9 addresses whether asynchronous release uses synchronous release sites within the active zone. The authors conclude that asynchronous release uses a different complement of sites than synchronous release, based on the partial localization of asynchronous release sites to synchronous release sites. The difficulty I have with this comparison is that the synchronous release sites define the sites, so they will cluster, by definition, making it a difficult comparison to the asynchronous events. I would suggest that maybe a complementary additional test could be to look at the distance between any synchronous or asynchronous event to the nearest neighboring synchronous event. The expectation is that synchronous events should have closer neighboring events than asynchronous events.

4. The 50 ms is a rather long window for defining synchronous release. This may be necessary to achieve adequate SNR for the localization, but I would expect significant contamination from asynchronous release in the "synchronous" component, which one would expect to result in occasional ectopic events in the synchronous signal. However, since all synchronous events are defined to be in the active zone, this would cause an artificial broadening of the apparent active zones. Does this confound the definition of the active zones?

5. The relationship between amplitude and location for synchronous release could reflect the timing of exocytosis. Events happening earlier in the frame will be integrated longer and thus brighter than the ones later in the frame. It could mean that the earliest events are near the middle of the active zone (opposite to their conclusion for the later events). The fact that EGTA-AM breaks down the relationship fits with this idea.

[Editors' note: further revisions were suggested prior to acceptance, as described below.]

Thank you for resubmitting your work entitled "Two forms of asynchronous release with distinctive spatiotemporal dynamics in central synapses" for further consideration by *eLife*. Your revised article has been evaluated by Lu Chen (Senior Editor) and a Reviewing Editor.

The manuscript has been improved but there are a few remaining issues that need to be addressed, as outlined below in the comments by Reviewer #3. I believe these issues can be largely addressed with some additional analysis and textual clarifications. We look forward to reading the revised manuscript.

*Reviewer #2 (Recommendations for the authors):*

I believe that the additional experiments as well as changes to the manuscript including the discussion made in this first round of revisions have greatly improved the manuscript. A great job by the authors that has satisfied all questions that I raised previously.

*Reviewer #3 (Recommendations for the authors):*

The authors have mostly addressed my concerns in the revision with their thorough responses to reviewer critiques, but the new data brings up a couple of new points for consideration regarding the relationship between event size and decay kinetics and endocytosis.

1. To address concerns about lateral diffusion, the authors decreased the temperature of the cultures to 25 degrees and noted that there was no change in spread or amplitude of the events. This was suggested as a means to specifically manipulate diffusion, yet endocytosis in many systems including some fast endocytosis in neurons (e.g. Delvendahl et al., 2016) is highly temperature dependent. This seems to run counter to the idea that event amplitude and kinetics are determined by endocytosis.

2. Related to point 1 above, the experiments with Dyngo (figure 5, supplement 1H) appear to show that blocking endocytosis decreases the amplitude of synchronous events. If endocytosis is driving the decay, then shouldn't events become larger?

3. What are the average event kinetics in dyngo? Since the authors have done the experiments, average traces showing the decay kinetics of individual events in dyngo vs control could be informative.

---

## [Author Response]

Reviewer #1 (Recommendations for the authors):1. A quantification of the individual synaptic preferences for different types of release would be informative – did some synapses primarily release vesicles via synchronous release, active zone asynchronous, or ectopic asynchronous release?

We thank the reviewer for this suggestion and now show the proportion of each type of release for all the recorded synapses (Figure 1E). The prevalence for synchronous release varied from ~40% to 92% in the synaptic population with an average of ~66% of all events being synchronous. 98% of synapses had asynchronous events inside the AZ. The relative preference for this type of release varied in the synapse population from ~0 to ~40% of all events, with an average of ~25%. ~79% of the synapses displayed ectopic asynchronous events, which varied from 0 to ~20% in synapse population, and on average represented ~9% of all events. We now have added this information in the Results, p. 6.

2. Since the authors already have the data, it would be interesting to quantify a nanocolumn phenotype at each active zone – specifically the number of active zones that display > 1 clusters of release sites such as in Tang et al. 2016.

We thank the reviewer for this point and now provide quantification of the number of release sites detected per AZ in Figure 1—figure supplement 1B. To make this analysis more comparable to Tang et al., 2016, we also performed this quantification with an additional requirement that a release site must undergo repeated reuse during our observation period (i.e. having 2 or more events) to be counted (Figure 1—figure supplement 1C). Within these definitions, we detected 8.9 ± 0.2 clusters/release sites per AZ or 2.8 ± 0.1 repeatedly reused release sites per AZ (with two or more events detected) (Figure 1—figure supplement 1B,C), in close agreement with the previous estimates Sakamoto et al.Tang et al., 2016( 2018; ). These results are now described in the text on p. 8.

3. What percentage of fluorescence decay is ultra-fast vs. fast – does the increased intercept for synchronous release mean a higher percentage of individual events go through calcium dependent endocytosis? Quantify the percentage for each type of release.

We thank the reviewer for this suggestion and now provide this quantification in Figure 5—figure supplement 1A and in Results, p.10. On average, for synchronous events, ultrafast and fast components represented ~58% and ~42% of total decay, respectively. For asynchronous events, this proportion was shifted towards smaller percentage of the fast component (~38% and ~33% for in-AZ and ectopic events, respectively), in agreement with our earlier observations.

4. Synchronous and asynchronous events having the same kinetics for ultra-fast endocytosis could be at limitation of the speed of reacidification and quenching of the fluorescence probe, preventing an observable difference in kinetics. Please discuss this in the discussion.

We agree, and have added this point in Discussion, p. 22.

5. Quantification of the amplitude of synchronous and asynchronous release at same distances from AZ center would be informative – I am assuming based on example traces that at the smallest distance synchronous and asynchronous are the same.

We appreciate this suggestion and now provide this quantification in Figure 5—figure supplement 1D,E. We quantified the correlation between event amplitude and distance to the AZ center using Pearson's linear correlation algorithm. In agreement with our earlier observations, this analysis revealed a strong (0.98) and highly significant (p = 0.015) correlation for synchronous release, while there was no significant correlation found for asynchronous release (p = 0.92) (Figure 5—figure supplement 1D), or in the presence of EGTA-AM (p = 0.25) (Figure 5—figure supplement 1E). Synchronous and asynchronous events had the same amplitude near the AZ center (the smallest distance bin), as the reviewer pointed out. These results are now described on p. 11.

6. The addition of more methodology details interspersed in the Results section as opposed to just citing previous work from the lab would help in ease of reading (it can be brief). i.e. – how were average traces made (averaged for the same bouton, one field of view, etc.), how were MVR events excluded from analysis?

We agree and have provided more details of the methodology in the Results, p. 8, 10, 12, 13 and in the Methods section, p25-28.

7. The addition of cartoons/graphics to make figures more legible would be helpful – i.e. in Figure 1 expanding Ai and Aii to include acquisition rate info, a stimulation protocol illustration, timeline of observation period, especially since a study design caveat is if all release sites were observed during the observation period

We thank the reviewer for this suggestion and have expanded the cartoons in Figure 1Ai/Aii to include the suggested details.

Reviewer #2 (Recommendations for the authors):1) The authors looked to investigate coupling of exo/endocytosis and clustered events in 100nm rings and observed a "gradual and significant increase in the amplitude of synchronous events from the center of the AZ towards periphery". It is unclear what the statistics are to differentiate this small difference. It seems possible given the lower temporal measurements of 50ms time frames makes these minor changes liable to differences in time of timing of vesicle fusion in relation to calcium sources. Is it possible that distance to a calcium source might change the timing of the synchronous phase of vesicle fusion?

We thank the reviewer for this important point and performed extensive additional analyses and new experiments to address it.

1. We quantified the correlation between event amplitude and distance to the AZ center using Pearson's Linear Correlation algorithm. We observed a strong (0.98) and highly significant (p = 0.015) correlation for synchronous release, while there was no significant correlation found for asynchronous release (p = 0.92) (Figure 5—figure supplement 1D), or in the presence of EGTA-AM (p = 0.25) (Figure 5—figure supplement 1E). This analysis provides strong evidence for the statistical significance of the observed distance-dependent effect.

2. Our results suggested that the observed larger average amplitude of synchronous vs asynchronous events can be explained by the differences in exo-/endocytosis coupling between the two forms of release. To further test this notion, we performed additional measurements of both types of release in the presence of dynamin inhibitor dyngo-4a. Dyngo-4a eliminated the differences in average event amplitude between synchronous and asynchronous events (Figure 5—figure supplement 1H), supporting the notion that these amplitude differences are endocytosis-dependent. Because the number of detected events was strongly reduced in the presence of dyngo-4a, we were unable to perform spatial analyses with this dataset. Nevertheless, this observation together with our earlier results suggests that the difference in the average event amplitude between the two forms of release is not only calcium-dependent (blocked by EGTA/Sr^2+^), but also dynamin-dependent, supporting the notion that it is a physiologically-relevant and endocytosis-dependent phenomenon.

3. We examined an additional hypothesis (also suggested by Reviewer 3), that confounds between the timing of fusion events and the number of photons collected could result in different spatial resolution of events in different locales of the AZ or between synchronous and asynchronous events, thus contributing to observed differences in amplitude. To examine this possibility, we determined the error of event localization as a function of distance and form of release. We found little variation between synchronous and asynchronous events (Figure 1—figure supplement 1A) or between events at the AZ center vs periphery (Figure 5—figure supplement 3A), and thus it cannot account for the observed differences in event amplitude.

4. We considered the possibility suggested by the reviewer that differences in the distance to a calcium source may account for the distance–dependent effect by changing the timing of the synchronous vesicle fusion at different locales of the AZ. We note that this mechanism does not negate the relevance of the observed effect, but rather provides a physiological basis for it. Since we observed that amplitude of synchronous events is larger at the AZ periphery, this would imply that vesicle fusion is closer to calcium sources and occurs earlier at the AZ periphery. While it is very difficult to directly prove or exclude this hypothesis, four lines of evidence argue against this being the predominant mechanism of the distance-dependent effect we observed:

(i) We found that EGTA eliminated the distance-dependent effect by reducing event amplitude at the periphery but not the AZ center. In contrast, we previously observed that release sites near the AZ center have several fold higher release probability and are affected stronger by EGTA than the release sites at the AZ periphery (Maschi and Klyachko, 2017). Thus the release sites near the AZ center are presumably closer to the calcium sources, which is opposite in direction of changes from the distance-dependent effect we observed here.

(ii) No differences in event amplitude were observed between synchronous and asynchronous events near the AZ center, although asynchronous events presumably occur farther away from the calcium sources (Kaeser and Regehr, 2014).

(iii) We found little variation in event localization error between synchronous and asynchronous events or between synchronous events at the AZ center vs periphery (Figure 1—figure supplement 1A and Figure 5—figure supplement 3A).

(iv) While an indirect argument, spatio-temporal properties of MVR also argue for preferential earlier vesicle fusion closer to the AZ center rather than periphery, which is opposite from the observed distance-dependent effect. Specifically, our previous analyses of MVR (Maschi and Klyachko, 2020) showed that one event within the MVR pair often has a larger amplitude and occurs earlier in time. This larger-amplitude event typically occurs closer to the AZ center, which is again opposite in direction from the distance-dependent effect observed here. These considerations have been added to Discussion in the revised version, p. 20-21.

2) In relation to the above comments, given the imaging window compared to acquisition, would this be changing observed peaks of fluorescence, that could translate into changes in magnitude as well as fitting for endocytosis?

We appreciate the reviewer’s concern. If we understood it correctly, this is a conceptually similar concern to the one also raised by the Reviewer 3, that the relationship between the timing of fusion events and the number of photons collected could result in different spatial resolution of events at the AZ center vs periphery or between synchronous and asynchronous events, thus contributing to observed differences in amplitude. As described above, to examine this possibility, we determined the error of event localization and found little variation between synchronous and asynchronous forms of release (Figure 1—figure supplement 1A) or between events at the AZ center vs periphery (Figure 5—figure supplement 3A), and thus it cannot account for the observed differences in event amplitude. Furthermore, we observed no measurable differences in amplitude between synchronous and asynchronous events near the AZ center, suggesting that variability in event timing, while present, is not the principal determinant of the event amplitude in our measurements. This is probably the case because of the limits set by the detection algorithm: the asynchronous events that are strongly delayed relative to the beginning of acquisition frame do not produce enough photons to reach the detection threshold and are rejected. These considerations have been added to Discussion on p. 19 and 20.

3) It seems that a limited number of hypothesis are tested and that the limited speed of imaging might be causing the differences observed. It is interesting that this difference is not seen in other conditions, asynchronous, Sr2+ and EGTA, but these are all altering the time locking of calcium-evoked fusion as well. Given the rather small nature of the difference that seems close to the St. Dev. of the noise as shown in Figure 3A and slow nature of imaging required by the probes, this part of the manuscript is less convincing.

We thank the reviewer for this point. We performed numerous additional experiments and analyses to provide further support for our initial results and to examine several additional hypotheses, as summarized below. Some of these experiments and analyses (Points #1,2,5 below) are described in detail in the response to Point 1 above, and are only briefly summarized here:

1. We improved quantification and statistical analysis of the relationship between event amplitude and distance to the AZ center using Pearson's Linear Correlation algorithm (Figure 5—figure supplement 1D,E). This analysis provides strong evidence for the statistical significance of the observed distance-dependent effect.

2. We performed experiments with a dynamin inhibitor dyngo-4a (Figure 5—figure supplement 1H), providing further support for the notion that amplitude differences between synchronous and asynchronous events are endocytosis-dependent.

3. We performed extensive additional experiments to improve temporal resolution in our measurements from 50ms to 25ms to achieve a more precise definition of synchronous vs asynchronous release using a near-TIRF imaging. We obtained essentially the same results on the differences in spatial organization of synchronous and asynchronous release, as we initially reported (Figure 1H and 1I). We also found that spatial differences between synchronous and asynchronous release events are largely the same for the three time intervals after the stimulus: 25-50ms, 50-75ms and 75-100ms (Figure 1H and 1I). However, because of the increased noise and reduced number of detected events in these measurements, we were unable to perform robust comparison of event amplitudes or their spatial dependences.

4. We performed additional experiments at room temperature to determine contribution from diffusion to event decay (Figure 5—figure supplement 3C-F) and also quantified the contribution from photobleaching (p. 14) to demonstrate that these factors cannot explain the observed differences in event amplitude.

5. We determined the error of event localization as a function of distance and type of release, and found little variation between synchronous and asynchronous forms of release (Figure 1—figure supplement 1A) or between events at the AZ center vs periphery (Figure 5—figure supplement 3A), and thus it cannot account for the observed differences in event amplitude.

Taken together, these results and analyses provide strong additional support for the physiological relevance of the distance-dependent effect observed, its endocytosis-dependent nature, and shows that this is unlikely an artifact of temporal limitations in out recordings or an effect of diffusion.

Reviewer #3 (Recommendations for the authors):The work is a careful analysis of individual vesicle fusion events visualized following stimulation. The authors use detailed analysis to parse the spatial and temporal features of fusion events within 50 ms of an AP (synchronous) and at later time points (asynchronous). While I think the findings are interesting, there are a few concerns that I feel the authors should address.1. I am concerned about the possible confounds between the timing of fusion events, the number of photons collected and the spatial resolution. In particular, the spatial resolution is known to be a function of the SNR for any particular event and thus dimmer events will result in poorer spatial precision in the measurements. The authors state that the stimulus is precisely timed to the beginning of a frame and since most synchronous events will occur soon after, they should integrate the signal over the entire frame. The asynchronous events may occur at any point during the frame and will thus be expected to have a dimmer peak intensity (figure 3F seems to confirm this difference) and presumably poorer spatial resolution. The relationship between distance to the center and the intensity in figure 3F seems to partially argue against this, but doesn't fully address whether some of the apparent differences in localization of asynchronous effects might result from differences in precision of localization (for example, the localization to release sites and the finding of ectopic events).

We thank the reviewer for this point and agree. To address this concern, we determined the error of event localization (i.e. spatial precision of localization) and found minimal variation between synchronous and asynchronous forms of release (Figure 1—figure supplement 1A), or between events at the AZ center vs periphery (Figure 5—figure supplement 3A), and thus it cannot account for the observed differences in event localization or their amplitudes. We also note that we observed no measurable differences in amplitude between synchronous and asynchronous events near the AZ center, suggesting that variability in event timing, while present, is not the principal determinant of the event amplitude or localization precision in our measurements. This is probably the case because of the limits set by the detection algorithm: the asynchronous events that are strongly delayed relative to the beginning of acquisition frame do not produce enough photons to reach the detection threshold and are rejected. These considerations have been added to Discussion on p. 19 -21.

Second, to further address this concern, we performed extensive additional experiments to improve temporal resolution in our measurements from 50ms to 25ms to achieve a more precise definition of synchronous vs asynchronous release, using a near-TIRF imaging (Figure 1H and 1I). We obtained essentially the same results as we initially reported, including two spatially distinct asynchronous event subpopulations, the presence of ~30% of ectopic asynchronous events, and the preferential spatial bias of asynchronous events inside the AZ towards the AZ center.

Together these results provide strong support for our initial observations and suggest that the observed spatial differences of synchronous and asynchronous events are not caused by differences in timing or precision of localization.

2. The analysis of endocytosis is performed by looking at the kinetic features of fluorescence decay following exocytosis. Other groups have found that approximately 20% of synaptic vesicle proteins escape the boutons by diffusion. To what extent is the endocytosis signal diffusion out of the ROI? Is there an estimate to the expected loss due to photobleaching?

We thank the reviewer for this point and considered the potential contribution from diffusion of vesicular components upon fusion, which could play a role in the observed spatial effect if contribution of diffusion is different at the AZ center vs periphery. First, we examined the average spatial profile of synchronous release events as a function of distance and found that it had a similar width near the AZ center vs periphery (Figure 5—figure supplement 3B), arguing against major spatial differences in diffusion across the AZ. This is consistent with the observation that asynchronous events do not have any measurable spatial differences in amplitude or decay kinetics and thus in the contribution from diffusion in different locations of the AZ, and diffusion presumably affects asynchronous and synchronous forms of release similarly.

Second, to further examine contribution from diffusion, we considered that the membrane diffusion coefficient decreases several fold with the decrease in temperature from 37°C to room temperature (Ries et al., 2009). In contrast, when we performed our measurements at room temperature, we found that the event’s spatial profile and its decay were nearly the same at the two temperatures (Figure 5—figure supplement 3C-F), and thus minimally affected by changes in diffusion with temperature. This observation supports the notion that diffusion does not play a major role in the observed differences in event amplitude or decay kinetics.

Finally, we quantified photobleaching and found that it accounted only for 3.3% of event decay during 1 sec interval, and thus also cannot account for differences in event decay between the two forms of release.

These results are now presented on p. 14.

3. The analysis on page 9 addresses whether asynchronous release uses synchronous release sites within the active zone. The authors conclude that asynchronous release uses a different complement of sites than synchronous release, based on the partial localization of asynchronous release sites to synchronous release sites. The difficulty I have with this comparison is that the synchronous release sites define the sites, so they will cluster, by definition, making it a difficult comparison to the asynchronous events. I would suggest that maybe a complementary additional test could be to look at the distance between any synchronous or asynchronous event to the nearest neighboring synchronous event. The expectation is that synchronous events should have closer neighboring events than asynchronous events.

We agree and have performed the suggested nearest neighbor analysis (Figure 3D). We observed that only ~20% of asynchronous events were localized within 50nm from a synchronous event, supporting our earlier findings.

4. The 50 ms is a rather long window for defining synchronous release. This may be necessary to achieve adequate SNR for the localization, but I would expect significant contamination from asynchronous release in the "synchronous" component, which one would expect to result in occasional ectopic events in the synchronous signal. However, since all synchronous events are defined to be in the active zone, this would cause an artificial broadening of the apparent active zones. Does this confound the definition of the active zones?

We thank the reviewer for this important point and agree. To address the reviewer’s concern we performed extensive additional experiments to improve temporal resolution of our measurements using a near-TIRF imaging of vGluT1-pHluorin-labeled vesicles to achieve an improved imaging rate of 25ms/frame. Accordingly, we were able to shorten the definition of synchronous release to within 25ms from the stimulus, and compare it to 3 temporal windows for asynchronous release (25-50ms, 50-75ms and 75-100ms). We observed essentially the same results as before, including the presence of ~35% of ectopic release among asynchronous events (Figure 1H), and the preferential bias of asynchronous events inside the AZ for the AZ center (Figure 1I). We also found that the properties of asynchronous release were very similar for the three temporal windows examined, with a slight increase in the proportion of ectopic vs “in-AZ” asynchronous events with time, from 33% to 37% to 39% for the three time windows, respectively. This result suggests that ectopic events are more likely to occur with longer time delays. The increased proportion of ectopic events (~33-39%) in these measurements as compared to 50ms/frame recordings (~27%) is consistent with the notion that the AZ dimensions are slightly broadened in 50ms recordings*.* Overall, this analysis demonstrates that the precise duration of the temporal windows used to define synchronous vs asynchronous release does not affect the observed core differences in the spatial organization of the two forms of release, at least at the timescales tested. These results are now presented on p. 7-8.

5. The relationship between amplitude and location for synchronous release could reflect the timing of exocytosis. Events happening earlier in the frame will be integrated longer and thus brighter than the ones later in the frame. It could mean that the earliest events are near the middle of the active zone (opposite to their conclusion for the later events). The fact that EGTA-AM breaks down the relationship fits with this idea.

We thank the Reviewer for this point, which was also raised by the Reviewer 2, and agree that event timing is an important factor to consider. We note that this mechanism does not negate the relevance of the observed effect, but rather provides a physiological basis for it. Since the amplitude of synchronous events is larger at the AZ periphery in our measurements (Figure 5F and Figure 5—figure supplement 1D), this hypothesis implies that synchronous vesicle fusion occurs earlier at the AZ periphery comparing to the AZ center. While it is a plausible mechanism, which is very difficult to directly prove or exclude, there are several lines of evidence that argue against this being the predominant mechanism of the distance-dependent effect we observed for the following reasons:

(i) No significant differences in amplitude were observed between synchronous and asynchronous events near the AZ center. This suggests that variability of event timing, while present, is not the key determinant of the detected event amplitude, at least in these measurements.

(ii) Ectopic events have significantly longer temporal delays comparing to asynchronous events inside the AZ (Figure 4A,B), but these two populations show no measurable differences in average event amplitude or decay kinetics (Figure 5—figure supplement 1B,C).

(iii) Our additional experiments showed that the difference in average amplitude between synchronous and asynchronous events is eliminated by inhibition of dynamin with dyngo-4a (Figure 5—figure supplement 1H), providing further support for the notion that the observed effect is an endocytosis-dependent phenomenon.

(iv) As described above, we found little variation in localization error between synchronous and asynchronous events (Figure 1—figure supplement 1A), or between events at the AZ center vs periphery (Figure 5—figure supplement 3A).

(v) While an indirect argument, spatiotemproal properties of multi-vesicular release (MVR) events argue for preferential earlier vesicle fusion near the AZ center rather than periphery, which is opposite from the observed distance-dependent effect. Specifically, our previous analyses of MVR (Maschi and Klyachko, 2020) showed that one event within the MVR pair often has a larger amplitude and occurs earlier in time. This larger-amplitude event typically occurs closer to the AZ center, which is opposite in the direction from the distance-dependent effect observed here.

These considerations are now presented in Discussion, p. 20-21.

[Editors' note: further revisions were suggested prior to acceptance, as described below.]

Reviewer #3 (Recommendations for the authors):The authors have mostly addressed my concerns in the revision with their thorough responses to reviewer critiques, but the new data brings up a couple of new points for consideration regarding the relationship between event size and decay kinetics and endocytosis.1. To address concerns about lateral diffusion, the authors decreased the temperature of the cultures to 25 degrees and noted that there was no change in spread or amplitude of the events. This was suggested as a means to specifically manipulate diffusion, yet endocytosis in many systems including some fast endocytosis in neurons (e.g. Delvendahl et al., 2016) is highly temperature dependent. This seems to run counter to the idea that event amplitude and kinetics are determined by endocytosis.

We thank the reviewer for this point and would like to clarify that it is caused by the lack of clarity in our description of these experiments.

We examined the temperature-dependence of endocytosis in our measurements, but failed to present this clearly in the previous version of the manuscript. While the membrane diffusion coefficient is expected to decrease ~10 fold with the decrease in temperature from 37°C to room temperature (Ries et al., 2009), we observed no significant changes in the kinetics of ultra-fast endocytosis (Figure 5figure supplement 3F) and only modest changes in the fast calcium-dependent component of endocytosis (Q_10_ ~1.3), suggesting that temperature-dependent changes in the event spatial profile (if present) can be attributed predominately to lateral diffusion. We note that the different magnitude of temperature-dependence of endocytosis we observed here as compared to (Delvendahl et al., 2016) could be related to the major differences in experimental conditions between our measurements (lowfrequency stimulation, 1Hz) and this previous study (50Hz trains). We and others (Pyott and Rosenmund, 2002) (Klyachko and Stevens, 2006) (Micheva and Smith, 2005) (Coulter et al., 1989) previously reported high temperature-dependence of synaptic facilitation and augmentation resulting from residual presynaptic calcium build up during high-frequency trains, which likely affected calciumdependent endocytosis during 50Hz trains in the above study, but not in our low-frequency experiments.

We also note that analysis of event spatial profile at room temperature was limited to the first 3 frames (0-150ms) only. This is nearly an order of magnitude faster timescale than the fast calcium-dependent form of endocytosis (~1s timescale), the component which we observed to differ for synchronous vs asynchronous release. Thus, lack of major changes in synchronous event profile that we observed in the first 3 frames at room temperature does not, in a large part, reflect this form of endocytosis. Finally, we also note that event amplitudes in our analysis were normalized to the peak in the first frame to allow comparison of spatial profiles between the two temperatures. Thus these amplitudes cannot be compared between conditions, as presented, and we made no conclusions about changes in event amplitudes with temperature.

We have now clarified these points in the Results, p. 14, and corresponding figure legend.

2. Related to point 1 above, the experiments with Dyngo (figure 5, supplement 1H) appear to show that blocking endocytosis decreases the amplitude of synchronous events. If endocytosis is driving the decay, then shouldn't events become larger?

We appreciate the reviewer’s concern. As in the point above, we believe it is caused by the lack of clarity in our description of these experiments. Multiple independent factors that influence the synchronous event amplitude in different conditions, including the prevalence of MVR and background florescence, in addition to endocytosis, change with application of dyngo-4a. Thus alterations in amplitudes of single synchronous events recorded in different conditions in different synapse populations cannot be interpreted to represent changes in endocytosis alone. We can only compare amplitudes of synchronous vs asynchronous events as these were recorded within the same synapses.

To address the reviewer’s concern, we presented pair-wise comparisons of synchronous to asynchronous event amplitudes in each condition as two separate graphs in Figure 5 figure supplement 1H.

3. What are the average event kinetics in dyngo? Since the authors have done the experiments, average traces showing the decay kinetics of individual events in dyngo vs control could be informative.

We had major difficulties performing and analyzing dyngo-4a experiments because it markedly reduced the Pr, resulting in greatly reduced number of observed events. Dyngo-4a also increased background noise presumably due to inhibition or delay of some forms of endocytosis resulting in larger proportion of vGluT1-pHluorin residing on the plasma membrane. As a result, we could not obtain reliable fits of single event traces to obtain data comparable to analyses shown in Figure 5D.

Additionally, while dynamin inhibitors, such as dyngo-4a, have been widely used to study endocytosis in synapses (Chung et al., 2010; Linares-Clemente et al., 2015; McCluskey et al., 2013; Morlot and Roux, 2013), recent studies suggested heterogeneity/complexity in the role of dynamins in vesicle endocytosis under different conditions (Afuwape et al., 2020). Therefore, we only included a single analysis in the presence of dyngo-4a comparing synchronous vs asynchronous event amplitude as a supplemental data. We strongly prefer not to include any further analyses of dyngo-4a in the current manuscript and we hope that the reviewer agrees.

References

Afuwape, O.A.T., Chanaday, N.L., Kasap, M., Monteggia, L.M., and Kavalali, E.T. (2020). Persistence of quantal synaptic vesicle recycling following dynamin depletion. bioRxiv, 2020.2006.2012.147975.

Chung, C., Barylko, B., Leitz, J., Liu, X., and Kavalali, E.T. (2010). Acute dynamin inhibition dissects synaptic vesicle recycling pathways that drive spontaneous and evoked neurotransmission. J Neurosci *30*, 1363-1376.

Coulter, D.A., Huguenard, J.R., and Prince, D.A. (1989). Calcium currents in rat thalamocortical relay neurones: kinetic properties of the transient, low-threshold current. J Physiol *414*, 587-604.

Delvendahl, I., Vyleta, N.P., von Gersdorff, H., and Hallermann, S. (2016). Fast, TemperatureSensitive and Clathrin-Independent Endocytosis at Central Synapses. Neuron *90*, 492-498.

Klyachko, V.A., and Stevens, C.F. (2006). Temperature-dependent shift of balance among the components of short-term plasticity in hippocampal synapses. The Journal of neuroscience : the official journal of the Society for Neuroscience *26*, 6945-6957.

Linares-Clemente, P., Rozas, J.L., Mircheski, J., Garcia-Junco-Clemente, P., Martinez-Lopez, J.A., Nieto-Gonzalez, J.L., Vazquez, M.E., Pintado, C.O., and Fernandez-Chacon, R. (2015). Different dynamin blockers interfere with distinct phases of synaptic endocytosis during stimulation in motoneurones. J Physiol *593*, 2867-2888.

McCluskey, A., Daniel, J.A., Hadzic, G., Chau, N., Clayton, E.L., Mariana, A., Whiting, A., Gorgani, N.N., Lloyd, J., Quan, A.*, et al.* (2013). Building a better dynasore: the dyngo compounds potently inhibit dynamin and endocytosis. Traffic *14*, 1272-1289.

Micheva, K.D., and Smith, S.J. (2005). Strong effects of subphysiological temperature on the function and plasticity of mammalian presynaptic terminals. The Journal of neuroscience : the official journal of the Society for Neuroscience *25*, 7481-7488.

Morlot, S., and Roux, A. (2013). Mechanics of dynamin-mediated membrane fission. Annual review of biophysics *42*, 629-649.

Pyott, S.J., and Rosenmund, C. (2002). The effects of temperature on vesicular supply and release in autaptic cultures of rat and mouse hippocampal neurons. J Physiol *539*, 523-535.

Ries, J., Chiantia, S., and Schwille, P. (2009). Accurate determination of membrane dynamics with line-scan FCS. Biophys J *96*, 1999-2008.